# Enhancing Cooperative Multi-Agent Reinforcement Learning with State Modelling and Adversarial Exploration

**Andreas Kontogiannis** [* 1 2]  **Konstantinos Papathanasiou** [* 3]  **Yi Shen** [4]
**Giorgos Stamou** [1]  **Michael M. Zavlanos** [4]  **George Vouros** [5]

## Abstract

Learning to cooperate in distributed partially observable environments with no communication abilities poses significant challenges for multi-agent deep reinforcement learning (MARL). This paper addresses key concerns in this domain, focusing on inferring state representations from individual agent observations and leveraging these representations to enhance agents' exploration and collaborative task execution policies. To this end, we propose a novel state modelling framework for cooperative MARL, where agents infer meaningful belief representations of the non-observable state, with respect to optimizing their own policies, while filtering redundant and less informative joint state information. Building upon this framework, we propose the MARL SMPE$^2$ algorithm. In SMPE$^2$, agents enhance their own policy's discriminative abilities under partial observability, explicitly by incorporating their beliefs into the policy network, and implicitly by adopting an adversarial type of exploration policies which encourages agents to discover novel, high-value states while improving the discriminative abilities of others. Experimentally, we show that SMPE$^2$ outperforms state-of-the-art MARL algorithms in complex fully cooperative tasks from the MPE, LBF, and RWARE benchmarks.

## 1. Introduction

**Background and Motivation.**  In cooperative multi-agent deep reinforcement learning (MARL), the goal is to train the learning agents in order to maximize their shared utility function. Many real-world applications can be modeled as fully cooperative MARL problems, including multi-robot cooperation (Alami et al., 1998), wireless network optimization (Lin et al., 2019), self-driving cars (Valiente et al., 2022), air traffic management (Kontogiannis & Vouros, 2023), and search & rescue (Rahman et al., 2022).

In most real-world applications, agents, acting in a decentralized setting, have access to the environment state through mere observations. However, collaborating under partial observability is a challenging problem, inherent from the non-stationary nature of the multi-agent systems. To address this problem, different approaches have been proposed; e.g., approaches which aim agents to model other agents (e.g., in (Hernandez-Leal et al., 2019)), to communicate effectively with others (e.g., in (Guan et al., 2022b)), or to leverage advanced replay memories (e.g., in (Yang et al., 2022)).

This paper draws motivation from *fully* cooperative MARL real-world applications (Papadopoulos et al., 2025) and focuses on the celebrated CTDE schema (Lowe et al., 2017a), where agents aim to learn effective collaborative policies. During training, agents share information about their observations, but in execution time they must solve the task given only their own local information. Specifically, we are interested in settings where agents lack explicit communication channels during execution. Such settings are of particular interest, because, while communication-based methods leverage inexpensive simulators for training, they may incur substantial computational overhead when executed in real-world environments (Zhu et al., 2022; Zhang & Lesser, 2013). This paper aims agents to infer meaningful beliefs about the unobserved state and leverage them for enhanced cooperation with others.

*Agent modelling* (AM) (Albrecht & Stone, 2017), also referred to as opponent modelling (He et al., 2016; Foerster et al., 2018a), has been proposed as a solution to infer beliefs about the global state, or the joint policy, under partial observability. Applications of AM in MARL have been extensively explored (Papoudakis et al., 2021; Raileanu et al., 2018; Sun et al., 2024; Hernandez-Leal et al., 2019; Nguyen et al., 2023; Tan & Chen, 2023; Gupta & Kahou,

---
[*]Equal contribution  [1]School of Electrical and Computer Engineering, NTUA, Greece [2]Archimedes, Athena Research Center, Greece [3]Department of Mathematics, ETH Zurich [4]Dept. of Mechanical Engineering & Materials Science, Duke University [5]Dept. of Digital Systems, University of Piraeus, Greece. Correspondence to: Andreas Kontogiannis <andreaskontogiannis@mail.ntua.gr>.

*Proceedings of the 42nd International Conference on Machine Learning*, Vancouver, Canada. PMLR 267, 2025. Copyright 2025 by the author(s).

2023; Fu et al., 2022), aiming agents to infer belief representations about other agents' policies or the unobserved state. Nonetheless, current AM approaches may pose challenges in enhancing agents' policies for the following reasons: (a) Standard AM may infer belief representations that are suboptimal for enhancing the agents' policies, because these representations are not learnt w.r.t. maximizing the agent's value function (e.g., in (Papoudakis et al., 2021; Papoudakis & Albrecht, 2020; Nguyen et al., 2023; Hernandez-Leal et al., 2019; Fu et al., 2022)). (b) Standard AM does not account for redundant, less informative, joint state features in learning the agent's beliefs (e.g., in (Raileanu et al., 2018; Hernandez-Leal et al., 2019; Papoudakis et al., 2021; Nguyen et al., 2023; Tan & Chen, 2023)), which has been shown to harm performance (Guan et al., 2022b). (c) AM representations are not leveraged in MARL to improve the agents' initial random exploration policies, thus being ineffective for enhancing policies in sparse-reward settings (as we show in Section 4.4). Moreover, AM approaches may be impractical because they typically involve a single learnable controller (e.g., in (Papoudakis et al., 2021; Nguyen et al., 2023; Fu et al., 2022)), or make strong assumptions regarding: the observation feature space; e.g., assuming a priori knowledge about what the observation features represent (Nguyen et al., 2023; Tan & Chen, 2023)), the nature of the game (e.g., being applicable only to team-game settings (Sun et al., 2024)), or settings where agents observe other agents' actions (Hu & Foerster, 2019; Hu et al., 2021; Fu et al., 2022) in execution time.

Taking the above significant challenges into account, this paper addresses the following research questions:

**Q1:** *Can agents learn to infer informative (latent) state representations given their own observations, in order to enhance their own policies towards coordination?*

**Q2:** *Can agents leverage the inferred state representations to efficiently explore the state space and discover even better policies?*

**High-level Intuition.** To understand the high-level intuition guiding our approach, imagine a fully cooperative task from the well-known LBF benchmark, where agents must first engage in extensive joint exploration to identify a specific food target. Once identified, all agents must then execute coordinated joint actions, simultaneously consuming the target. At each time step, each agent—*based on its current local information*—attempts to infer unobserved yet informative state information (e.g., the food target's location and the positions of other agents with whom it must cooperate). The agent then refines its policy using informative state inferences accumulated *throughout the entire trajectory*. To enhance *joint exploration*, each agent is further incentivized to implicitly discover novel observations, thereby improving

the state inference capabilities of other agents and ultimately facilitating better cooperation.

**Main Contributions.** **(a)** We propose a novel *state modelling* framework for cooperative MARL under partial observability. In this framework, agents are trained to infer meaningful state belief representations of the non-observable joint state w.r.t. optimizing their own policies. The framework assumes that the joint state information can be redundant and needs to be appropriately filtered in order to be informative to agents. Also, the framework entails multi-agent learning and does not impose assumptions on either what the observation features represent, or access to other agents' information during execution. **(b)** Building upon the state modelling framework, we propose "**S**tate **M**odelling for **P**olicy **E**nhancement through **E**xploration"[1] (SMPE[2]), a MARL method which aims to enhance agents' individual policies' discriminative abilities under partial observability. This is done explicitly by incorporating their beliefs into the policy networks, and implicitly by adopting adversarial exploration policies aiming to discover novel, high-value states while improving the discriminative abilities of others. More specifically, SMPE[2] leverages (i) *variational inference* for inferring meaningful state beliefs, combined with *self-supervised learning* for filtering non-informative joint state information, and (ii) intrinsic rewards to encourage *adversarial exploration*. **(c)** Experimentally, we show that SMPE[2] significantly outperforms state-of-the-art (SOTA) MARL algorithms in complex fully cooperative tasks of the Multiagent Particle Environment (MPE) (Mordatch & Abbeel, 2018; Lowe et al., 2017b), Level-Based Foraging (LBF) (Albrecht & Stone, 2017) and the Multi-Robot Warehouse (RWARE) (Papoudakis et al., 2020) benchmarks, several tasks of which have been highlighted as open challenges by prior work (Papadopoulos et al., 2025).

## 2. Preliminaries

### 2.1. Cooperative MARL as a Dec-POMDP

A Dec-POMDP (Oliehoek et al., 2016) for an $N$-agent cooperative task is a tuple $(S, A, P, r, F, O, N, \gamma)$, where $S$ is the state space, $A$ is the joint action space $A = A_1 \times \cdots \times A_N$, where $A_i$ is the action space of agent $i$, $P(s' \mid s, a) : S \times A \to [0, 1]$ is the state transition function, $r(s, a) : S \times A \to \mathbb{R}$ is the reward function and $\gamma \in [0, 1)$ is the discount factor. Assuming partial observability, each agent at time step $t$ does not have access to the full state, yet it samples observations $o_t^i \in O_i$ according to the observation function $F_i(s) : S \to O_i$. Agents' joint observations are denoted by $o \in O$ and are sampled according to $F = \prod_i F_i$. The action-observation history

---

[1]Our official source code can be found at https://github.com/ddaedalus/smpe.

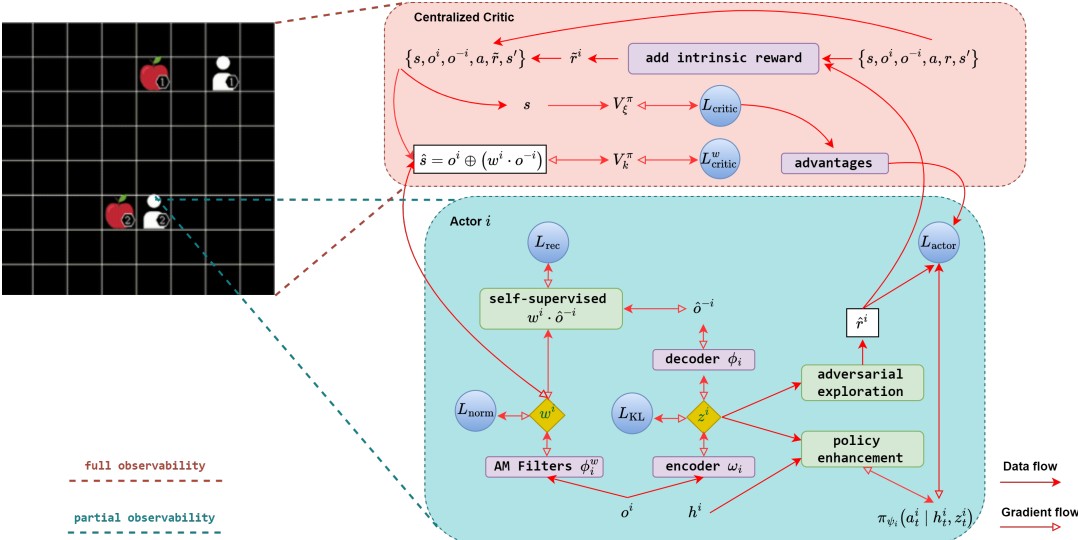

*Figure 1.* Overview of SMPE$^2$: SMPE$^2$ is built on top of the MAA2C algorithm (Papoudakis et al., 2020). Each agent's actor (*blue box*) has partial observability. Conditioned on agent' own observation $o^i$, it reconstructs other agents' observations $o^{-i}$ using a variational encoder-decoder (ED) which infers a belief state embedding $z^i$. To filter non-informative joint state information and learn only from informative features, the actor utilizes agent modelling (AM) filters ($w^i$) which filter the reconstruction targets of ED in a self-supervised learning manner. Moreover, the embeddings $z^i$ are exploited to provide intrinsic rewards $\hat{r}^i$, in order to facilitate the proposed adversarial exploration schema. Finally, the actor exploits the belief embeddings $z^i$ to enhance the policy $\pi_\psi(a_t^i \mid h_t^i, z_t^i)$. Concurrently, SMPE$^2$ leverages two critic models (*red box*), namely $V_\xi(s)$ and $V_k(\hat{s})$ (parameterized by $\xi$ and $k$ accordingly), both accessing full observability. $V_\xi$ is the standard critic of backbone MAA2C and $V_k(\hat{s})$ approximates explicitly the joint value function at modified states $\hat{s}$ purified by the AM filters $w^i$. During execution agents do not have access to others' observations, but use the inferred $z^i$ to discriminate effectively between different states despite partial observability. The full algorithm of SMPE$^2$ can be found in Algorithm 1 (Appendix).

for agent $i$ at time $t$ is denoted by $h_t^i \in H_i$, which includes action-observation pairs until $t$-1 and $o_t^i$, on which the agent can condition its individual stochastic policy $\pi_{\theta_i}^i(a_t^i \mid h_t^i) : H_i \times A_i \to [0,1]$, parameterised by $\theta_i$. The joint action of all agents other than $i$ is denoted by $a^{-i}$, and we use a similar convention for the policies, i.e., $\pi_\theta^{-i}$. The joint policy is denoted by $\pi_\theta$, with parameters $\theta \in \Theta$.

The objective is to find an optimal joint policy which satisfies the optimal value function $V^* = \max_\theta \mathbb{E}_{a\sim\pi_\theta, s'\sim P(\cdot|s,a), o\sim F(s)} \left[ \sum_{t=0}^{\infty} \gamma^t r_t \right]$.

### 2.2. The state modelling optimization framework

The main idea behind the proposed state modelling (SM) optimization framework is to allow agents, *conditioned only on their own observations*, to infer meaningful beliefs of the unobserved state which would help them optimize their individual policies. Our state modelling framework connects state-representation learning with policy optimization.

Formally, we consider that each agent $i$ aims to learn a model of the non-observable joint state: The model infers a probability distribution over beliefs regarding the non-observable joint state, based on agents' local observations. The model is parameterized by $\omega_i$ and is denoted by $p_{\omega_i}$.

State modelling assumes a belief space $\mathcal{Z}_i$ of latent variables $z^i$, such that $z^i \sim p_{\omega_i}(\cdot \mid o^i)$ contains meaningful information about the unobserved state, informative for optimizing the agent's own policy. In particular, we assume that states contain information redundant for optimising agents' individual policies. Specifically, we consider a joint state feature to be *non-informative* for agent $i$ if:

- it is irrelevant to agent $i$ for maximizing its future rewards (if assuming that the agent had access to it)

- it cannot be inferred through $z^i$, in the sense that the agent cannot predict it conditioned on its own observation due to partial observability and non-stationarity.

Considering the above assumptions, the state modelling objective is to find meaningful latent beliefs $z^i$ *with respect to optimizing* the joint policy $\pi_\psi$, resulting from individual policies $\pi_{\psi_i}$ parameterized by $\psi_i$, i.e. $\pi_{\psi_i}(a_t^i \mid h_t^i, z_t^i) : H_i \times \mathcal{Z}_i \times A_i \to [0,1]$, as follows:

$$V_{SM}^* = \max_\omega \max_\psi V_{SM}(\omega, \psi)$$

$$= \max_{\omega,\psi} \mathbb{E}_{z\sim p_\omega} \left[ \mathbb{E}_{a\sim\pi_\psi, s'\sim P(\cdot|s,a), o\sim F(s)} \left[ \sum_{t=0}^{\infty} \gamma^t r_t \right] \right] \tag{1}$$

where $V_{SM}^*$ is the optimal value function under SM.

**Proposition 2.1.** *The state modelling objective equals the DecPOMDP objective; i.e. $V_{SM}^* = V^*$.*

Although both objectives are intractable, Proposition 2.1 implies that $V_{SM}$ allows us to explore how agents can form meaningful beliefs $z^i$ about the unobserved states with sufficient state-discriminating abilities to enhance their own policies without constraining the candidate policies, i.e. those that utilize $z^i$, to be suboptimal w.r.t. the value function.

## 3. State Modelling for Policy Enhancement through Exploration (SMPE$^2$)

We propose a state modelling MARL method, called SMPE$^2$. The method can be separated into two distinct, but concurrent, parts. The first part (Section 3.1) involves self-supervised state modelling, where each agent learns a model to infer meaningful belief state representations based on own and other agents' observations. The second part (Section 3.2) involves the explicit use of the inferred representations to encourage agents towards an adversarial type of multi-agent exploration through intrinsic motivation. The learning process of SMPE$^2$ is illustrated in Figure 1.

### 3.1. Self-supervised state modelling

To learn a representation of the unobserved state, we aim agents to learn the reconstruction of informative features of other agents' observations using only their own observations. The reconstruction aims at inferring latent beliefs, $z^i$, about the unobserved state, assuming that joint observations provide sufficient evidence to discriminate between states. SMPE$^2$ uses amortized variational inference (Kingma & Welling, 2013), i.e., the reconstruction model of agent $i$ is a probabilistic encoder ($q_{\omega_i}$)-decoder ($p_{\phi_i}$) (ED). One would assume that ideally ED should perfectly predict ($\hat{o}^{-i}$) other agents' observations ($o^{-i}$), so that $z^i$ could be as informative about what other agents observe as possible. However, as it was highlighted in prior work (Guan et al., 2022b), using the full state information as an extra input to the policy, even when utilizing a compressed embedding, may harm performance due to redundant state information non-informative to agent $i$.

To mitigate the use of redundant state information, SMPE$^2$ filters out non-informative features of $o^{-i}$, i.e., state features that are irrelevant to agent $i$ for maximizing its future rewards and cannot be inferred through $z^i$. To achieve this, we introduce *agent modeling (AM) filters*, denoted by $w_j^i$, which serve as learnable weight parameters—one for each of the other agents $j \in -i$. SMPE$^2$ utilizes $w_j^i = \sigma(\phi_i^w(o^j))$, where $\phi_i^w$ is parameterized by an MLP, and $\sigma$ is the sigmoid activation function, ensuring that the filter values satisfy $w_j^i \in [0, 1]$. Intuitively, $w^i$ has an AM interpretation, as it

represents the importance of each of other agents' information to agent $i$'s state modelling.

Formally, for each agent $i$, SMPE$^2$ considers a modification of the state modelling framework of (1) to the following maximization problem:

$$\underset{\omega_i, \phi_i, \psi_i}{\text{maximize}} \quad V_{SM}(\omega, \psi) + \lambda \cdot \text{ELBO}(\omega_i, \phi_i; o^i, w^i \cdot o^{-i}) \quad (2)$$

where $\cdot$ (in $w^i \cdot o^{-i}$) denotes element-wise multiplication and we define $\text{ELBO}(\omega_i, \phi_i ; o^i, w^i \cdot o^{-i})$ to be equal to

$$\mathbb{E}_{z^i \sim q_{\omega_i}} \left[ \log p_{\phi_i}(w^i \cdot o^{-i} \mid z^i) \right] - \text{KL} \left( q_{\omega_i}(z^i \mid o^i) \parallel p(z^i) \right)$$

that is, the Evidence Lower bound (Kingma & Welling, 2013) which uses $o^{-i}$ filtered by $w^i$ as the reconstruction targets.

Although problem (2) is also intractable, the ELBO term would allow SMPE$^2$ to find good solutions: ED aims to identify informative observation features in the reconstruction, through the AM filters, leading to $z^i$ which enhance the individual agents' policies towards cooperation. In doing so, SMPE$^2$ entails enhanced performance of the joint policy by *finding state beliefs $z^i$ w.r.t maximizing the value function*. This comes in contrast to other approaches (e.g., (Papoudakis et al., 2021; Papoudakis & Albrecht, 2020; Nguyen et al., 2023; Hernandez-Leal et al., 2019; Fu et al., 2022)) which may suffer from distribution mismatch (Ma et al.), as $z^i$ is disconnected from the policy. We note that $\lambda$ in (2) is a hyperparameter ensuring that the ELBO term does not dominate the state modelling objective $V_{SM}$.

Based on (2), in the reconstruction part of SMPE$^2$, ED has as prediction target the other agents' observations, $o_t^{-i}$ filtered by $w^i$. More specifically, the ED per agent $i$ minimizes the following *self-supervised* reconstruction loss (using the reparameterization trick (Kingma & Welling, 2013), with $\omega_i$ being Gaussian):

$$L_{\text{rec}}(\omega_i, \phi_i, \phi_i^w) = \| \widetilde{w}^i \cdot o^{-i} - w^i \cdot \hat{o}^{-i} \|^2 \quad (3)$$

where we use targets $\widetilde{w}^i$ to filter the target observations $o^{-i}$ in order to stabilize the training of $w^i$.

Both $(\omega_i, \phi_i)$ and $w^i$ are learned in a self-supervised manner, since the targets in Equation (3) are explicitly factorized by the AM filters. Note the importance of the AM filters $w^i$: (a) With it, although the target of ED grows linearly with the number of agents, only features that can be inferred through $z^i$ remain as part of other agents' observations in the reconstruction loss. (b) Without it, it would be challenging to infer meaningful embeddings $z^i$, due to non-informative joint state information. Thus, $w^i$ explicitly controls the reconstruction of the initial other agents' observations, since it adjusts how much ED would penalize the loss for each target feature.

However, if both $w^i$ and $\tilde{w}^i$ are equal to 0, then these assignments are solutions to (3). To ensure that $w^i$ does not vanish to zero throughout all dimensions, we add the following regularization loss:

$$L_{\text{norm}}(\phi_i^w) = -\|w^i\|_2^2 \qquad (4)$$

To make the embeddings $z^i$ variational (and thus probabilistic as in the definition of state modelling), following the problem in (2), we add the standard KL divergence regularization term, as follows:

$$L_{\text{KL}}(\omega_i) = \text{KL}\big(q_{\omega_i}(z^i|o^i) \,\|\, p(z^i)\big), \quad p(z^i) = \text{N}(0, I) \qquad (5)$$

As for the policy optimization part, SMPE$^2$ can be implemented using any actor-critic MARL algorithm. We implement SMPE$^2$ using MAA2C (see Appendix C.1), due to its good performance on various benchmarks (Papoudakis et al., 2020; Papadopoulos et al., 2025) and its efficient natural on-policy learning. Following the definition of the state modelling problem, we aim to ensure that $w^i$ (and thus $z^i$) incorporate information relevant to maximizing $V^\pi$ and thus, $w^i$ to be capable of filtering non-informative state features irrelevant to maximizing agent's future rewards. To do so, alongside MAA2C's standard critic (parameterized by $\xi$), we train an additional critic (parameterized by $k$) which exploits $w^i$. Thus, we minimize the following losses:

$$
\begin{aligned}
L_{\text{critic}}(\xi) &= \left[ r_t^i + V_{\xi'}^\pi(s_{t+1}) - V_\xi^\pi(s_t) \right]^2 \\
L_{\text{critic}}^w(\phi_i^w, k) &= \left[ r_t^i + V_{k'}^\pi(\hat{s}_{t+1}) - V_k^\pi(\hat{s}_t) \right]^2
\end{aligned}
\qquad (6)
$$

where $\hat{s} = o^i \oplus (w^i \cdot o^{-i})$ is the filtered state from the view of agent $i$'s modelling, with $\oplus$ meaning vector concatenation, and superscript for target network. Overall, the loss for learning the state belief representations for agent $i$ is:

$$L_{\text{encodings}} = L_{\text{critic}}^w + \lambda_{\text{rec}} \cdot L_{\text{rec}} + \lambda_{\text{norm}} \cdot L_{\text{norm}} + \lambda_{\text{KL}} \cdot L_{\text{KL}} \quad (7)$$

where $\lambda_{\text{rec}}$, $\lambda_{\text{norm}}$ and $\lambda_{\text{KL}}$ are hyperparameters weighting the corresponding loss terms. The actor of each agent $i$ uses policy $\pi \equiv \pi_{\psi_i}$ enhanced by $z^i$. Thus, the actor loss is:

$$
\begin{aligned}
L_{\text{actor}}(\psi_i) = &-\beta_H \cdot H\left(\pi_{\psi_i}(a_t^i \mid h_t^i, z_t^i)\right) \\
&- \log \pi_{\psi_i}(a_t^i \mid h_t^i, z_t^i) \cdot \left(r_t^i + V_{\xi'}^\pi(s_{t+1}) - V_\xi^\pi(s_t)\right)
\end{aligned}
$$

Therefore, the overall SMPE loss is as follows:

$$L_{\text{SMPE}} = L_{\text{actor}} + L_{\text{critic}} + L_{\text{encodings}}.$$

## 3.2. Adversarial Count-based Intrinsic Exploration

We further empower the agents' policies to reach novel, high-value states, even in sparse-reward settings. To do so, we harness the rich state information captured by the belief $z^i$ and design intrinsic rewards which naturally account

for both individual and collective benefit. Specifically, our exploration strategy encourages each agent $i$ to effectively explore its own belief space so that it discovers novel $z^i$. To do so, given that $z^i$ is solely conditioned on $o^i$, the agent is implicitly motivated to discover novel observations that must lead to novel $z^i$. Crucially, our exploration schema implies an ***adversarial exploration*** framework fostering cooperation: Agent $i$ is intrinsically motivated to discover novel $o^i$ (which lead to novel $z^i$) which at the same time constitute unseen ED prediction targets for the others' reconstruction training. Therefore, these targets aim to adversarially increase the losses of other agents' reconstruction models. By doing do, agent $i$, except only for searching for novel $o^i$, implicitly strives to assign adversarial targets to other agents $-i$, to help them form better beliefs about the global state. In the same way, since $z^i$ is informative of both $o^i$ and $o^{-i}$, agent $i$, leveraging $o^{-i}$, is also urged to form a well-explored belief about what others observe, enhancing cooperation.

To leverage the above benefits, we adopt a simple *count-based* intrinsic reward schema based on the SimHash algorithm, similar to (Tang et al., 2017), but instead of hashing the agent's observations, we hash $z^i \in \mathcal{Z}$. Specifically, we utilize the SimHash function $SH : \mathcal{Z} \to \mathbb{Z}$ and calculate the intrinsic reward $\hat{r}^i$ as follows: $\hat{r}^i = 1/\sqrt{n(SH(z^i))}$, where $n(SH(z^i))$ represents the counts of $SH(z^i)$ for agent $i$. Agent $i$ uses the modified reward $\tilde{r}_t^i = (r_t^i + \beta \hat{r}_t^i)$, where hyperparameter $\beta$ controls the contribution of the intrinsic reward.

The choice of $SH$ is due to the fact that it allows nearby $z^i$ to be transformed into nearby hash values at low computational cost. A key note here is that the domain of the $SH$ function is dynamic. To make intrinsic rewards more stable and ensure the assumption of i.i.d. training data, $z^i$ becomes more stable by avoiding large changes in the ED parameters between successive training episodes. To do so, similarly to (Papoudakis et al., 2021), we perform periodic hard updates of $\omega_i$ and $\phi_i$ using a fixed number of training steps as an update period.

*Remark* 3.1. For the interested reader, we demonstrate in Appendix E.4.7 that the intrinsic rewards do not fluctuate; instead, they decrease smoothly throughout training until they reach a minimal plateau, as expected.

**Why is ED conditioned only on $o^i$?** We choose ED to be conditioned solely on $o^i$, instead of $h^i$. In particular, since $z^i$ is leveraged for intrinsic exploration, we make $z^i$ solely conditioned on $o^i$, so that the agent will be intrinsically rewarded more if the current observation was novel and led to novel $z^i$. On the other hand, if ED was indeed conditioned on $h^i$, then the intrinsic exploration would be less meaningful, since the main goal now would be to discover *novel trajectories instead of novel observations* (that lead to

novel $z^i$), thus making the novelty of the current observation *excessively compressed* within $z^i$. For instance, if the agent discovered a novel, high-value observation (thus contained in a high-value global state) within a well-explored trajectory, then the intrinsic reward would be quite low (because the whole trajectory would not be novel) and thus possibly unable to help the agent find a better policy. The above conceptual motivation does not defeat the assumption of partial observability, because the policy network is indeed conditioned on $h_t^i$ and therefore implicitly leverages a belief representation conditioned on all time steps, that is all per-time-step beliefs $z_t^i$.

## 4. Experimental Setup

In our experimental setup, we evaluate MARL algorithms on three widely recognized benchmarks: MPE (Mordatch & Abbeel, 2018; Lowe et al., 2017b), LBF (Albrecht & Stone, 2017), and RWARE (Papoudakis et al., 2020). Due to space constraints, technical details for these benchmarks can be found in Appendices E.1, E.2, and E.3, respectively. These benchmarks often require substantial coordination and exploration capabilities in order to discover effective joint policies and have been used to evaluate MARL algorithms in many related works, including (Papadopoulos et al., 2025; Christianos et al., 2021; Papoudakis et al., 2021; Guan et al., 2022a; Yang et al., 2022; Papoudakis et al., 2020).

Regarding the questions stated in the introduction, to address **Q1**, we first verify the effectiveness of the inferred state modelling embeddings of the proposed method on fully cooperative dense-reward tasks of the MPE benchmark. To this aim, in MPE we evaluate our method without using the proposed intrinsic rewarding schema. To answer **Q2**, we evaluate the effectiveness of full SMPE$^2$ on complex, fully cooperative, sparse-reward tasks of the LBF and RWARE benchmarks. In MPE we compare our method with MAA2C, COMA (Foerster et al., 2018b), MAPPO (Yu et al., 2022) and the transformer-based ATM (Yang et al., 2022). In LBF and RWARE, we also include the SOTA intrinsic motivation based methods: EOI (Jiang & Lu, 2021), EMC (Zheng et al., 2021) and MASER (Jeon et al., 2022), as they cover a wide range of different approaches for exploration in MARL; namely through diversity, curiosity and sub-goal generation, respectively.

Results are averaged over six random seeds, and the primary metric is the unnormalized average episodic reward. We report 25-75% confidence intervals (as in (Zheng et al., 2021)). Following (Papoudakis et al., 2020), we set the total number of steps to 10M for MPE and LBF, and 40M for RWARE.

### 4.1. Results on the MPE benchmark

In this section, we evaluate the proposed method without intrinsic rewards for exploration (SMPE) on two fully cooperative MPE tasks: *Spread* and *Double Speaker-Listener*. To assess scalability, we examine four scenarios with 3, 4, 5, and 8 agents in the *Spread* task, following a setup similar to (Ruan et al., 2022). Due to space limitations, results for *Double Speaker-Listener* are provided in Appendix B. As depicted in Figure 2, SMPE demonstrates superior performance in *Spread*, showing significant improvements in overall performance. Notably, as the number of agents and landmarks increases, SMPE consistently outperforms other methods by enabling agents to form informative beliefs about the unobserved state. This allows agents to maintain awareness of available landmarks even with increased complexity, a capability not easily achieved by other methods.

### 4.2. Results on the LBF benchmark

In this section, we evaluate the full SMPE$^2$ on six fully cooperative LBF tasks. As Figure 3 shows, SMPE$^2$ achieves superior performance compared to the other SOTA methods. In contrast to the other methods, SMPE$^2$ manages to fully solve challenging sparse-reward LBF tasks, including *2s-9x9-3p-2f* and *4s-11x11-3p-2f*, both of which have been highlighted as open challenges by prior work (Papadopoulos et al., 2025). As for the other LBF tasks, SMPE$^2$ either converges faster (a) to an optimal policy, or (b) to a good policy, or constantly achieves better episodic reward over time, even with a large grid (see *7s-20x20-5p-3f*).

SMPE$^2$'s success stems from its alignment with LBF's need for precise coordination among agents (through accurate beliefs about the unobserved state and good exploration policies) to collect all foods optimally. On the other hand, EMC and MASER completely fail to improve the well-known poor performance of QMIX (Rashid et al., 2020) in LBF (e.g., see (Papadopoulos et al., 2025; Papoudakis et al., 2020)), because both methods highly rely on the initial random policies and, as a result, they generate misleading intrinsic rewards based on low-value episodic data or irrelevant sub-goals, respectively. The only exception on this is the *2s-12x12-2p-2f* task, only because this task is the least sparse and finding a good, non-optimal policy is not that hard. EOI, which is built on top of MAA2C, does not perform as well as MAA2C, because it encourages the agents to explore through diversity, which may be unnecessary to the homogeneous agents in LBF.

### 4.3. Results on the RWARE benchmark

We evaluate SMPE$^2$ on three hard, fully cooperative RWARE tasks; namely *tiny-2ag-hard*, *tiny-4ag-hard* and *small-4ag-hard*. As shown in Figure 4, our SMPE$^2$ achieves

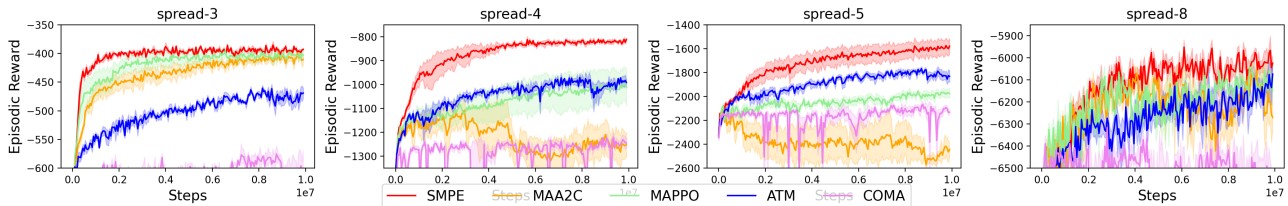

*Figure 2.* Results on the MPE benchmark

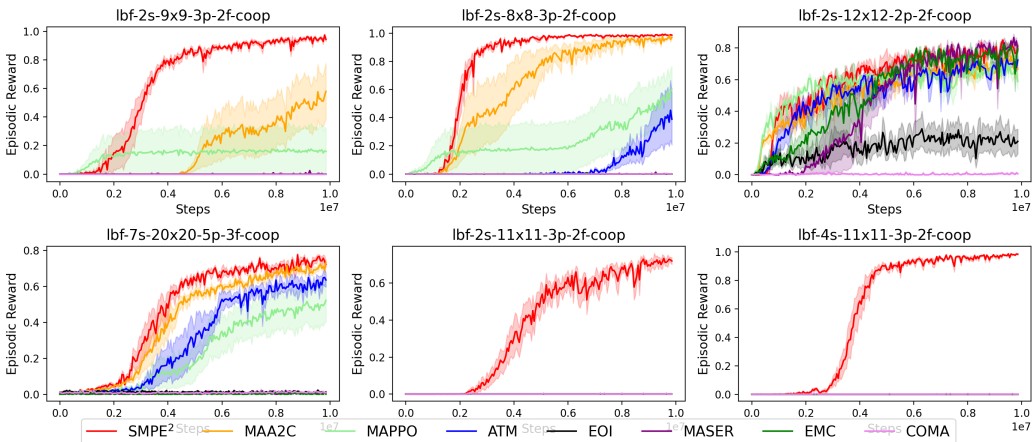

*Figure 3.* Results on the LBF benchmark

superior performance compared to the SOTA methods across all three tasks. These RWARE tasks are particularly challenging because they require: (1) effective exploration strategies to deal with sparse rewards; that is only very specific sequences of actions result to positive rewards (first load a specific shelf and then unload into an empty shelf), (2) agents to coordinate so that they avoid monopolizing all tasks individually or adopting policies that obstruct others, particularly in narrow passages, and (3) accurate modelling of the state due to excessive partial observability.

The failure of MASER and EMC is attributed to the first of the reasons. EOI displays good performance because it encourages the agents to be more diverse, thus some of them are able to find good policies. However, due to the second reason it completely fails in *small-4ag-hard*, because of insufficient coordination among agents. In contrast, SMPE$^2$ outperforms the SOTA methods, as: It addresses the third reason through modelling explicitly the joint state and also the first and second reasons through the effectiveness of the adversarial exploration schema.

### 4.4. Ablation Study

In our ablation study, we address the following questions:

**Q3:** *How important are state modelling, the AM filters and the adversarial exploration schema?*

**Q4:** *How expressive is the state modelling embedding $z^i$?*

**Q5:** *Does SMPE$^2$ outperform other modelling methods?*

**Q6:** *Is SMPE$^2$ flexible? Can it be combined with other MARL backbone algorithms?* (see Appendix E.4.9)

**Q7:** *Do we really need a second critic for training $w^i$ w.r.t. policy optimization?* (See Appendix E.4.3)

First, we address question **Q3**. Figure 5 validates the component selection in SMPE$^2$. It depicts (up) the importance of the AM filters ($w^i$) in better episodic reward and convergence to good policies and also the superiority of the proposed adversarial exploration schema over the standard method (Tang et al., 2017). Here, we note that the complete failure of *SMPE (no_intr)* is attributed to the fact that although agents are empowered by state modelling abilities, it may be difficult to adopt good exploration policies and thus to reach high-value states. Figure 5 (up) highlights that both excessive cooperation and good exploration are required to achieve good performance, as the baseline SMPE methods do not perform well. Thus, in these tasks, we would not have good exploration without good state modelling for effective collaboration, neither good collaboration without good exploration. Moreover, Figure 5 (down) demonstrates

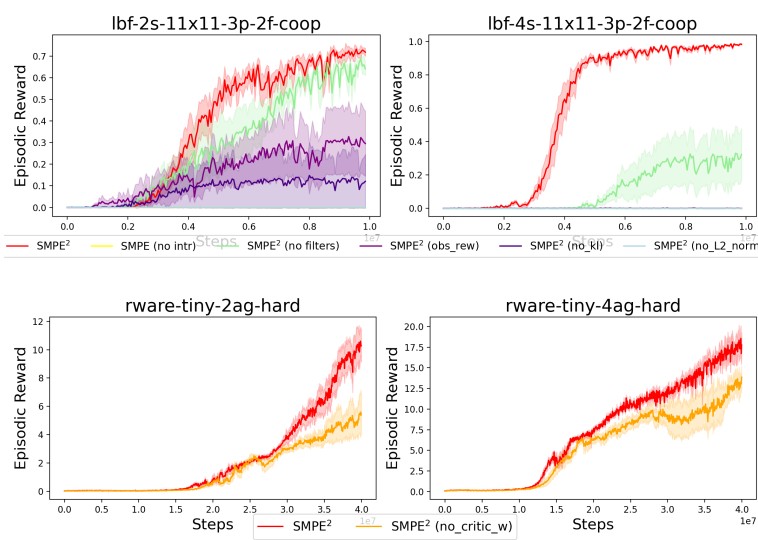

*Figure 4.* Results on the RWARE benchmark

*Figure 5.* Ablation study: **(up)** Validating component selection: SMPE$^2$ without intrinsic rewards (*no_intr*), AM filters (*no_filters*), $L_{\text{KL}}$ (*no_kl*), $L_{\text{norm}}$ (*no_L2_norm*), and with standard SimHash (Tang et al., 2017) replacing our exploration schema (*obs_rew*). **(down)** Learning $w^i$ (and thus $z^i$) w.r.t. policy optimization: we use SMPE$^2$ without $L_{\text{critic}}^w$ (*no_critic_w*) as baseline.

the impact of learning state representations w.r.t. policy optimization (as proposed in (1)), which can yield significant improvements in episodic reward and policy convergence.

To address **Q4**, in Figure 7, we present t-SNE representations (commonly used in MARL as in (Papoudakis et al., 2021; Xu et al., 2023; Liu et al., 2024)) of the state modelling beliefs $z^i$ of the three agents in LBF *2s-11x11-3p-2f* task. The figure demonstrates that when $L_{\text{KL}}$ is enabled, agents form cohesive beliefs: Their beliefs coincide in regions covering a large area, while maintaining individual beliefs in regions covering a smaller area. Conversely, without $L_{\text{KL}}$, agents' beliefs show less coherence, resulting in poorer joint exploration, as depicted in Figure 5. Table 1 (Appendix E.4.8) further validates these findings.

To address question **Q5**, we compare SMPE$^2$ to SIDE (Xu et al., 2022) and a multi-agent extension of the popular AM method LIAM (Papoudakis et al., 2021) (denoted by MLIAM). More details about MLIAM can be found in Appendix C.2. Figure 6 illustrates the comparison in MPE and

LBF tasks (benchmarks where LIAM was also originally tested). SMPE$^2$ significantly outperforms the baselines due to the following reasons: (a) MLIAM's objective is exacerbated by modelling the actions of others, leading to an inference problem with an exponentially large number of candidate target actions, (b) SIDE does not use the inferred $z^i$ in execution, (c) both methods do not account for non-informative state information, and (d) both methods do not leverage the inferred representations to improve joint exploration.

**MARL and Attention Modules.** MARL approaches (e.g., (Sunehag et al., 2018; Rashid et al., 2020; Son et al., 2019)) represent the joint state-action value as a function of individual models learnt based on the global reward. Attention modules (Vaswani et al., 2017) have been proposed: to encourage the cooperation of agents via agent-centric mechanisms (Shang et al., 2021), to estimate the value functions for explorative interaction among the agents (Ma et al., 2021), as advanced replay memories (Yang et al., 2022),

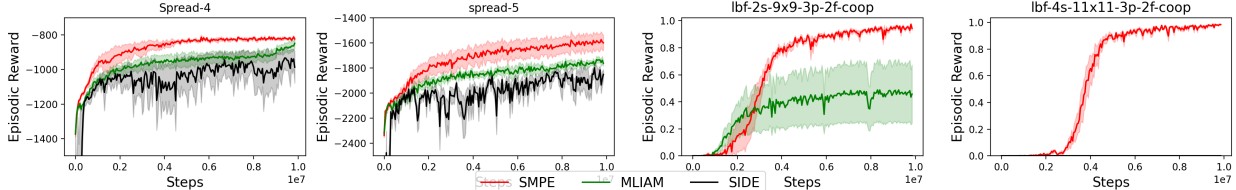

*Figure 6.* SMPE$^2$ against a custom implementation of the multi-agent extension of LIAM (Papoudakis et al., 2021) and SIDE (Xu et al., 2022)

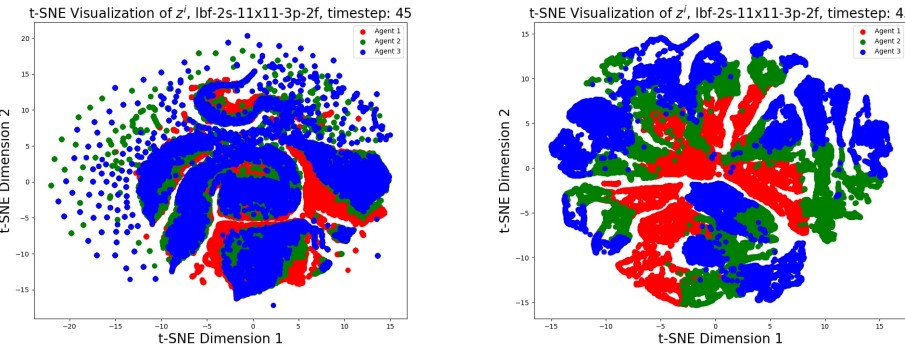

*Figure 7.* t-SNE visualization of the state modelling embedding $z^i$ (for the three agents; agent 1: red, agent 2: green, agent 3: blue) in an LBF task at the 45th time step; **(left)**: SMPE$^2$, **(right)**: *SMPE$^2$ (no_kl)* for ablation study with $L_{\mathrm{KL}} = 0$.

for efficient credit assignment (Zhao et al., 2023b), and for learning good policies (Ma et al., 2023; Wen et al., 2022; Zhao et al., 2023a).

**Agent Modelling (AM).** AM (Albrecht & Stone, 2017) has been proposed to model the policies of other agents or the unobserved state. Bayesian inference has been used to learn representations of probabilistic beliefs (Zintgraf et al., 2021; Moreno et al., 2021; Zhai et al., 2023), or even for explicitly modelling concepts derived from the theory of mind (Nguyen et al., 2020; Moreno et al., 2021; Hu et al., 2021). LIAM (Papoudakis et al., 2021) uses an ED to learn embeddings, by reconstructing the local information of other agents, and add them as extra input for policy learning. A key distinction between standard AM (Hernandez-Leal et al., 2019; Papoudakis et al., 2021; Nguyen et al., 2023) and our state modelling lies in their utilization: the former serves as an auxiliary task disconnected from policy optimization, while the latter does not necessitate an optimal full reconstruction, but only good reconstruction of the filtered targets to find good policies. Also, AM typically involves a single controller interacting with non-learnable agents (Hernandez-Leal et al., 2019; Papoudakis & Albrecht, 2020; Papoudakis et al., 2021; Nguyen et al., 2023; Fu et al., 2022), or makes strong assumptions regarding prior knowledge of observation features (Tan & Chen, 2023; Nguyen et al., 2023), access to other agents' information during execution

(Yuan et al., 2022; Hu & Foerster, 2019; Hu et al., 2021; Fu et al., 2022), or considers only team-game settings (Sun et al., 2024). Additionally, approaches such as (Raileanu et al., 2018; Hernandez-Leal et al., 2019; Nguyen et al., 2023; Tan & Chen, 2023; Xu et al., 2022) do not consider non-informative joint state information in learning beliefs, which can detrimentally affect performance (Guan et al., 2022b).

## 5. Conclusion and Future Directions

In this paper, we studied the problem of inferring informative state representations under partial observability and using them for better exploration in cooperative MARL. We proposed a state modelling optimization framework, on top of which we proposed a novel MARL method, namely SMPE$^2$. Experimentally, we demonstrated that SMPE$^2$ outperforms state-of-the-art methods in complex MPE, LBF and RWARE tasks.

In future work, we are interested in the following open challenges: (a) Can we further improve state modelling by incorporating transformers into the architecture? (b) How can we use state modelling to improve scalability in MARL? (c) Does state modelling adapt well in stochastic settings (i.e. settings with noisy observations, or more complicated dynamics)?

## Acknowledgements

This research was supported in part by project MIS 5154714 of the National Recovery and Resilience Plan Greece 2.0 funded by the European Union under the NextGenerationEU Program.

## Impact Statement

Multi-Agent Deep Reinforcement Learning (MARL) holds promise for a wide array of applications spanning robotic warehouses, search&rescue, autonomous vehicles, software agents, and video games, among others. Partial observability in these settings is an inherent feature as much as decentralization: This imposes challenges for the coordination and cooperation of agents that this work addresses. Indeed, this paper contributes the state modelling framework and SMPE$^2$ algorithm to the advancement of MARL models for such applications, in large and continuous state-action spaces, close to real-world problems. However, it is essential to acknowledge that the practical implementation of MARL methods faces significant challenges, including issues of limited theoretical guarantees, poor generalization to unseen tasks, explainability, legal and ethical considerations. These challenges, although beyond the immediate scope of our work, underscore the necessity for prioritizing extensive research in MARL. The overarching objective is to develop MARL agents capable of operating safely and addressing real-world problems effectively. It is imperative that many MARL methods undergo rigorous testing before deployment. While the potential benefits of safe, accurate, and cost-effective MARL applications are substantial, including the reduction of human effort in demanding tasks and the enhancement of safety, achieving these outcomes requires meticulous validation and refinement of the underlying methodologies.

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

## A. Further Related Work

**MARL and Exploration.** In sparse rewards settings, MARL exploration strategies encompass density-based (Tang et al., 2017; Zhao et al., 2023c; Jo et al., 2024), curiosity-driven (Zheng et al., 2021; Li et al., 2024), and information-theoretic (Li et al., 2021; Jiang & Lu, 2021) approaches. MAVEN (Mahajan et al., 2019) aims agents to explore temporally extended coordinated strategies. However, it does not encourage exploration of novel states and the inference of the latent variable still needs to explore the space of agents' joint behaviors. (Nguyen et al., 2023), which only considers a single learnable controller setting, employs social intrinsic motivation based on AM but relies on a priori knowledge of observation feature representations to compute intrinsic rewards. In contrast, SMPE$^2$ uses the informative state embeddings to encourage an adversarial type of multi-agent exploration through a simple intrinsic reward method.

## B. Results on MPE's Double Speaker Listerer

Due to page limit, we present the results on Double Speaker Listener in this section of the Appendix. As can be clearly shown, SMPE outperforms the other evaluated methods resulting in better joint policies. Interestingly, SMPE consistently demonstrates average episodic reward better than the other methods and converges to optimal policies.

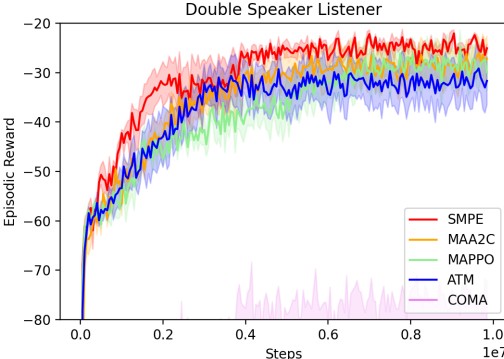

*Figure 8.* Results on Double Speaker Listener

## C. Extended Preliminaries

### C.1. Multi-Agent Actor Critic (MAA2C): Independent Actors with Centralized Critic

We consider the CTDE version of the MAA2C algorithm with independent actors and centralized critic, which has been widely used in MARL literature (Papoudakis et al., 2020; Xiao et al., 2022; Christianos et al., 2020; Su et al., 2021). The basic idea is that MAA2C constitues a simple extension of the single agent A2C algorithm (Mnih et al., 2016) to the CTDE schema. The actor network of agent $i$, parameterized by $\theta_i^\pi$, is conditioned only on the local observation history $h_t^i$. The critic network, whose parameters we denote by $\xi$, uses the full state, $s_t$, of the environment and approximates the joint value function $V^\pi$. The actor network is optimized by minimizing the actor-critic policy gradient (PG) loss:

$$L_{\text{actor}}(\theta_i^\pi) = -\log \pi_{\theta_i}(a_t^i \mid h_t^i) \cdot \left(r_t^i + \gamma V^\pi(s_{t+1}; \xi') - V^\pi(s_t; \xi)\right) - \beta_H H(\pi_{\theta_i}(a_t^i \mid h_t^i)) \tag{8}$$

where $H$ is the traditional entropy term of policy gradient methods. The critic network is trained by minimizing the temporal difference (TD) loss, calculated as follows:

$$L_{\text{critic}}(k) = \left(V^\pi(s_t; k) - y^i\right)^2 \tag{9}$$

where $y^i = r_t^i + \gamma V^\pi(s_{t+1}; k)$ is the TD target. We note that both the actor and the critic networks are trained on-policy trajectories.

### C.2. MLIAM: Multi-Agent Local Information Agent Modelling (LIAM)

In the original paper (Papoudakis et al., 2021), LIAM is an agent modelling method controlling one agent $i$ and assuming that the rest $-i$ modelled agents take actions based on some fixed set of policies $\Pi$. MLIAM is our custom implementation for the *multi-agent extension* of the single-agent method, LIAM: MLIAM assumes that all agents are learnable and homogeneous (i.e. they share their policy parameters). MLIAM's algorithm is similar to LIAM's, with the only exception that the following now hold for every agent, instead of only the single controlled agent. It assumes the existence of a latent space $\mathcal{Z}$ which contains information regarding the policies of the other agents and the dynamics of the environment. Aiming to relate the trajectories of each agent $i$ to the trajectories of the other agents $-i$ it utilizes an encoder-decoder framework. Specifically, at a timestep $t$ the recurrent encoder $f_w^i : \tau^i \to \mathcal{Z}$ takes as input the observation and action trajectory of the agent $i$ until that timestep $(o_{1:t}^i, \alpha_{1:t-1}^i)$ and produces an embedding $z_t^i$. Then the decoder $f_u^i : \mathcal{Z} \to \tau^{-i}$ takes the embedding $z_t^i$ and reconstructs the other agents' observation and action $(o_t^{-i}, \alpha_t^{-i})$. The encoder-decoder loss of agent $i$ for a horizon $H$ is defined as:

$$L_{\text{ED}}^i = \frac{1}{H} \sum_{t=1}^{H} [(f_u^{i,o}(z_t^i) - o_t^{-i})^2 - \log f_u^{i,\pi}(\alpha_t^{-i} \mid z_t^i)] \tag{10}$$

where $z_t^i = f_w^i(o_{:t}^i, \alpha_{:t-1}^i)$ and $f_u^{i,o}$, $f_u^{i,\pi}$ denote the decoders' output heads for the observations and actions respectively.

The learned embedding is used as an additional input in both the actor and the critic considering an augmented space $\mathcal{O}_{aug}^i = \mathcal{O}^i \times \mathcal{Z}$, where the $\mathcal{O}^i$ is agents' original observation space and $\mathcal{Z}$ is the embedding space. Considering an A2C reinforcement learning algorithm and a batch of trajectories $B$ the A2C loss of agent $i$ is defined as:

$$L_{\text{A2C}}^i = \mathbf{E}_{(o_t, a_t, r_{t+1}, o_{t+1} \sim B)} [\frac{1}{2}(r_{t+1}^i + \gamma V_\phi(s_{t+1}) - V_\phi(s_t))^2 - \hat{A} \log \pi_\theta(a_t^i \mid o_t^i, z_t^i) - \beta_H H(\pi_\theta(a_t^i \mid o_t^i, z_t^i))] \tag{11}$$

where $H$ is the entropy and $\beta$ a fixed hyperparameter controlling the initial random policy.

## D. Missing Proof

**Proof of Proposition 2.1**

*Proof.* Let $\pi_\psi$ be the joint policy parameterized by the joint parameters $\psi \in \Psi$. Then, we have:

$$V_{SM}^* = \max_\omega \max_\psi \mathbb{E}_{z \sim p_\omega} \left[ \mathbb{E}_{a \sim \pi_\psi, s' \sim P(\cdot|s,a), o=F(s)} \left[ \sum_{t=0}^\infty \gamma^t r_t \right] \right]$$

$$\geq \max_\psi \mathbb{E}_{a \sim \pi_\theta, s' \sim P(\cdot|s,a), o=F(s)} \left[ \sum_{t=0}^\infty \gamma^t r_t \right]$$

$$\geq \max_\theta \mathbb{E}_{a \sim \pi_\theta, s' \sim P(\cdot|s,a), o=F(s)} \left[ \sum_{t=0}^\infty \gamma^t r_t \right]$$

$$= V^*$$

where the inequalities hold because $\Theta \subset \Psi$ and thus every $\pi_\theta$ solving the Dec-POMDP problem, could be equivalent with some $\pi_\psi$ (e.g., when $\pi_\psi$ completely ignores the latent variable $z \sim p_\omega$). On the other hand, $V^*$ is the optimal state value function and thus it holds that $V_{SM}^* \leq V^*$ which implies that $V_{SM}^* = V^*$.

$\square$

# E. Extended Experimental Setup

The selected tasks from all three evaluated benchmark environments, namely MPE, LBF and RWARE, are partially observable and rely on the history of the observations of the learning agents.

## E.1. Multi-agent Particle Environment (MPE)

We used the source code in https://github.com/semitable/multiagent-particle-envs (under the MIT licence).

### E.1.1. SPREAD: A COOPERATIVE NAVIGATION TASK

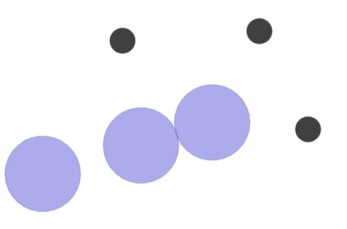

*Figure 9.* Spread: A cooperative navigation task

This fully cooperative environment involves $N$ agents and $N$ landmarks, with the primary goal of enabling agents to learn how to cover all landmarks while avoiding collisions. A key challenge is balancing global and local incentives: agents are rewarded based on the proximity of the closest agent to each landmark on a global scale, while local penalties are incurred for collisions. This structure creates a trade-off, requiring agents to coordinate effectively to minimize distances to landmarks while also avoiding collisions in a shared space. The dual challenge of learning collision avoidance and optimal landmark coverage in a multi-agent context demands the development of cooperative strategies, making this task particularly complex for multi-agent reinforcement learning (MARL), especially as $N$ increases. Insights derived from this environment have practical applications in traffic management systems, distributed sensor networks, urban planning, and smart cities. The environment is depicted in Figure 9.

- **Observation space**: Agents receive their current velocity, position, and the distance between landmarks and other

agent positions as their input.

- **Observation size**: [24,]

- **Action space**: The action space is discrete and involves 5 actions: standing still, moving right, left, up and down.

- **Reward**: All agents receive the same team reward, which includes the summed negative minimum distance to any other agent. Additionally, the collisions between any two agents are with a reward of -1.

In our experiments, the scenario *spread-n* means Spread with $n$ agents and $n$ landmarks.

### E.1.2. DOUBLE SPEAKER-LISTENER: A COOPERATIVE COMMUNICATION TASK

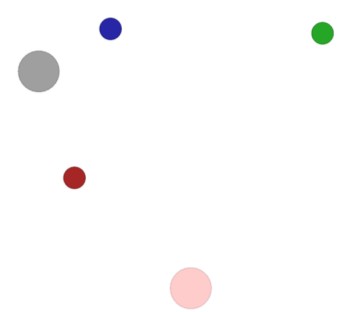

*Figure 10.* Double Speaker-Listener: A cooperative communication task

This fully cooperative environment features a Speaker and a Listener, where the Speaker is responsible for effectively communicating information about a target landmark while the Listener navigates based on the guidance provided by the Speaker's signals. Operating in a partially observable setting, this environment introduces challenges such as communication complexity and the need for extensive coordination despite limited perspectives and feedback. The insights derived from this environment have valuable applications in robotic teams, human-robot collaboration, and autonomous navigation systems. The environment is depicted in Figure 10.

- **Observation Space**: The speaker agent observes only the colour of the goal landmark which is represented as three numeric values. The listener agent receives as observations its velocity, relative landmark positions as well as the communication of the speaking agent.

- **Observation size**: [11,]

- **Action Space**: Similar to other MPE environments the listener's action is space is discrete and includes five actions: (standing still, moving right, left, up and down). The speaker's action space is also discrete however it includes three actions to communicate the goal to the listener.

- **Reward**: The reward is common for both agents and it is calculated as the negative square Euclidean distance between the listener's position and the goal landmark's position.

### E.2. Level-based Foraging (LBF)

We used the source code in https://github.com/uoe-agents/lb-foraging (under the MIT licence).

The Level-Based Foraging (LBF) (Albrecht & Stone, 2017) benchmark offers fully cooperative environments where agents navigate a two-dimensional grid-world to collect food items. Each agent and food item are assigned a specific level, and agents can move in four directions on the grid, collecting food only when the combined levels of the agents meet or exceed that of the food item. A significant challenge within the LBF environments is the sparse reward structure, as agents must coordinate effectively to simultaneously consume specific food items. The insights derived from this environment have

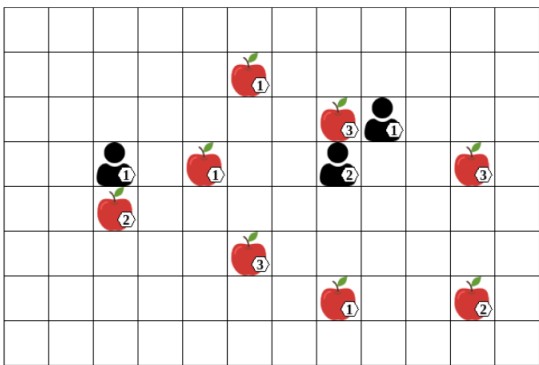

*Figure 11.* Level-based Foraging

practical applications in multi-robot collaboration, resource management in supply chains, and coordination during disaster response efforts. The environment is depicted in Section 4.2.

- **Observation space**: All agents receive triplets of the form $(x, y, level)$ as observations. The number of triples is equal to the sum of the number of foods and the number of agents in the environment. The observations begin with the food triples and are followed by the agents triples. Each triplet contains the x and y coordinates and level of each food item or agent.

- **Observation size**: $3 \times$ (number of agents + number of foods)

- **Action space**: The action space is discrete and involves six actions: standing still, move North, move South, move West, move East, pickup.

- **Reward**: In Level-Based Foraging environment agents are rewarded only when they pick up food. This reward depends on both the level of the collected food and the level of each contributing agent, and it is defined for the agent $i$ as follows:

$$r^i = \frac{\text{FoodLevel} * \text{AgentLevel}}{\sum \text{FoodLevels} \sum \text{LoadingAgentsLevel}}$$

Also rewards are normalized in order the sum of all agent's returns on a solved episode to equal one.

Last but not least, the LBF task under the name of "Ss-GxG-Pp-Ff-coop" corresponds to the *cooperative* task that has: an GxG grid consisting of $P$ agents, $F$ foods and with partial observability within a window of length $S$ for each agent.

*Remark* E.1. *2s-11x11-3p-2f* and *4s-11x11-3p-2f* are the most difficult task among the selected LBF ones, because: (a) a high-value state associated with some positive reward is practically impossible to be reached through an initial random policy (because a unique combination of agents must participate in the collection of specific food), and (b) selecting actions that lead to such states is very difficult due to the excessive partial observability of this task. Our results totally agree with the above facts, as no other method manages to find valuable states in the 11x11 tasks, thus resulting only in zero episodic reward over time.

### E.3. Multi-Robot Warehouse (RWARE)

We used the source code in https://github.com/uoe-agents/robotic-warehouse (under the MIT licence).

The Multi-Robot Warehouse (RWARE) (Papoudakis et al., 2020) environment models a fully cooperative, partially observable grid-world in which robots are tasked with locating, delivering, and returning requested shelves to workstations. Agents navigate within a grid that limits their visibility, providing only partial information about nearby agents and shelves, which makes their decision-making processes more complex. One of the primary challenges in this environment is the sparse reward structure; agents receive rewards only upon successfully delivering shelves, requiring them to follow a specific sequence of actions without immediate feedback. This situation demands effective exploration and coordination among

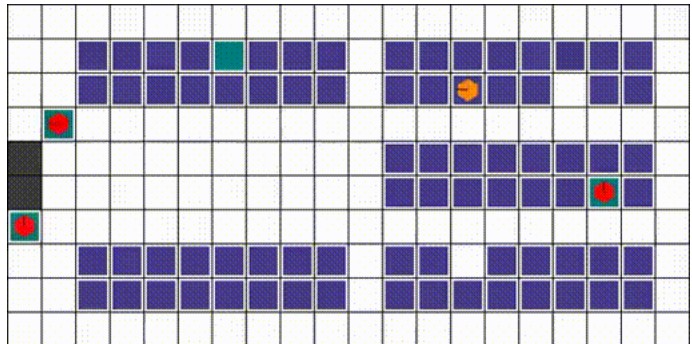

*Figure 12.* Multi-Robot Warehouse (RWARE)

agents to achieve their objectives. The insights gained from RWARE have meaningful applications in logistics and warehouse management, where multiple autonomous robots must work together to optimize inventory handling and order fulfillment. The tasks in RWARE vary in the world size, the number of agents, and shelf requests. The default size settings in the Multi-Robot Warehouse environment have four options: "tiny," "small," "medium," and "large." These size specify the number of rows and columns for groups of shelves in the warehouse. In the default setup, each group of shelves comprises 16 shelves arranged in an 8x2 grid. The difficulty parameter specifies the total number of requests at a time which by default equals the number of agents (N). One can modify the difficulty level by setting the number of requests to half ("-hard") or double ("-easy") the number of agents. The sparsity of rewards and high-dimensional sparse observations make RWARE a challenging environment for agents, as they need to perform a series of actions correctly before receiving any rewards. The environment is depicted in Figure 12.

- **Observation space**: The environment is partially observable, and agents can only observe a 3x3 grid containing information about surrounding agents and shelves. Specifically the observations include the agent's position, rotation, and whether it is carrying a shelf, the location and rotation of other robots, the shelves and whether they are currently in the request queue.

- **Observation size**: [71,]

- **Action space**: The action space is discrete and involves five actions: turn left, turn right, forward, load/unload shelf.

- **Reward**: The agents receive rewards only when they successfully complete the delivery of shelves.

## E.4. Extended Ablation Study

### E.4.1. COMPARISON OF SMPE$^2$ TO SMPE(*no_intr*)

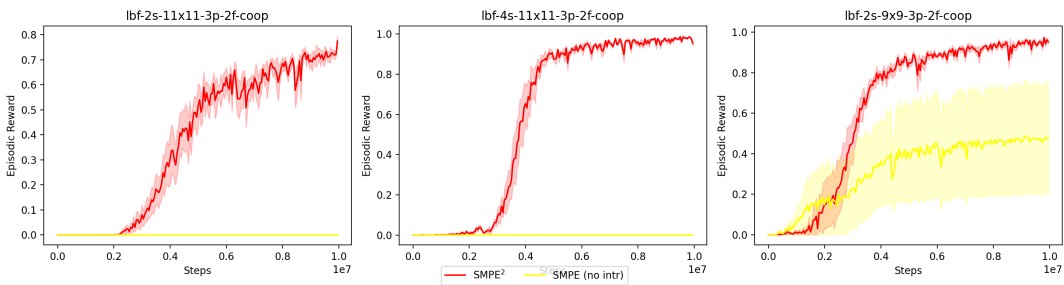

*Figure 13.* Comparison of SMPE$^2$ to SMPE(*no_intr*) on the three most difficult evaluated LBF tasks

### E.4.2. FURTHER RESULTS ON THE COMPARISON OF SMPE$^2$ TO MLIAM

In this subsection of the Appendix, we provide more results on the comparison of our SMPE$^2$ with MLIAM in Figure 14.

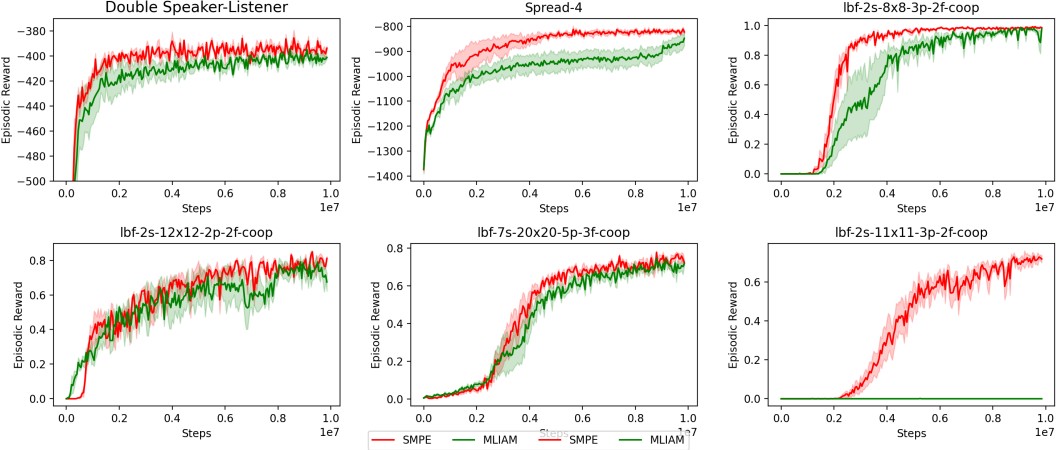

*Figure 14.* Further ablation study results on the comparison between SMPE and MLIAM on MPE and LBF

### E.4.3. WHY DO WE NEED A SECOND CRITIC FOR TRAINING $w^i$ W.R.T. POLICY OPTIMIZATION?

In this subsection of the Appendix, we study why SMPE$^2$ really needs two critics: one ($\xi$) for providing the advantages needed for training the actor and one ($k$) needed for training $w^i$ with respect to policy optimization. As can be clearly seen from Figure 15, the selection of two critic models is essential for the training of SMPE$^2$, as the baseline (*no_standard_critic*) which uses only the critic $k$ for both purposes displays both worse average episodic reward and convergence. More specifically, it suffers from high variance, attributed to the fact that in some runs it entirely diverges from good policies leading to very poor performance.

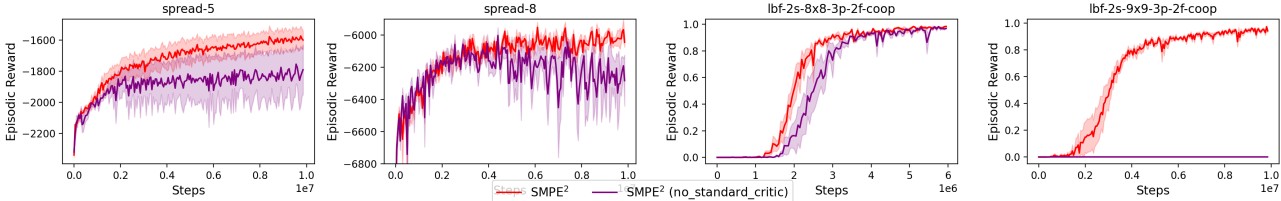

*Figure 15.* Ablation study on why we need the second critic model: Baseline SMPE$^2$ (*no_standard_critic*) is our method using the critic $k$, that is the one conditioned on $w^i$, also for providing the advantages needed to train the actor.

### E.4.4. ABLATION STUDY ON $\lambda_{\text{REC}}$

Figure 16 illustrates the importance of $\lambda_{\text{rec}}$ to the contribution of the reconstruction loss to the overall objective in a Spread task. As can be clearly seen from the plot, lowering the value of $\lambda_{\text{rec}}$ worsens both convergence and average episodic reward in this task. We attribute this to the fact that lowering the value of $\lambda_{\text{rec}}$ leads to worse state modelling, and thus the agents are not capable of inferring good beliefs about the unobserved state. Interestingly, as we increase the value of $\lambda_{\text{rec}}$, the overall performance deteriorates slightly, while displaying higher variance. This is because the reconstruction term is penalized more in the overall objective, dominating the policy optimization term.

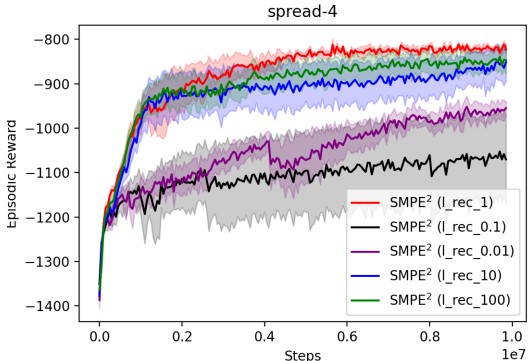

*Figure 16.* Ablation study on $\lambda_{\text{rec}}$

### E.4.5. ABLATION STUDY ON $\lambda_{\text{NORM}}$

Regarding $\lambda_{norm}$, in the ablation study we showed that if it equals $0$, then it can negatively affect the algorithm performance, because, a feasible solution for $w^i$ is to converge to $0$. Here, we provide a further ablation study in Figure 17, which shows that if $\lambda_{norm}$ is set to a value ranging from $[0.1, 1]$, then performance is pretty much the same.

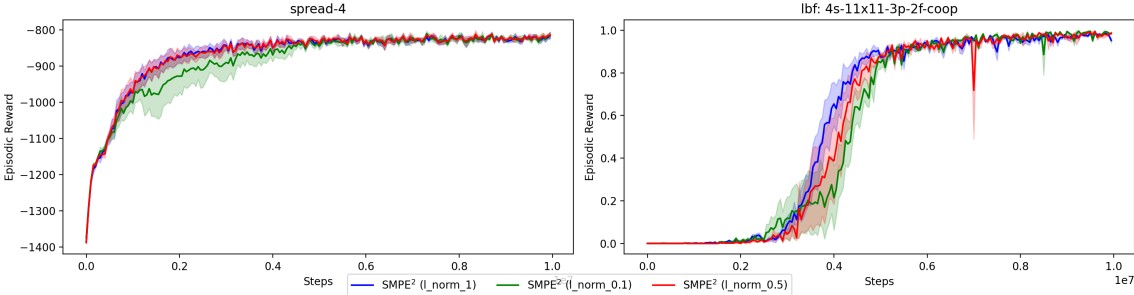

*Figure 17.* Ablation study on $\lambda_{\text{norm}}$

### E.4.6. RESULTS ON LEARNING THE STATE BELIEF REPRESENTATIONS WITH RESPECT TO POLICY OPTIMIZATION

In this subsection of the Appendix, we provide more results on the significance to learn the state belief representations with respect to policy optimization in Figure 18. As we discussed in Section 4.4, learning the state belief representations with respect to policy optimization can yield significant improvements in terms of average episodic reward and convergence.

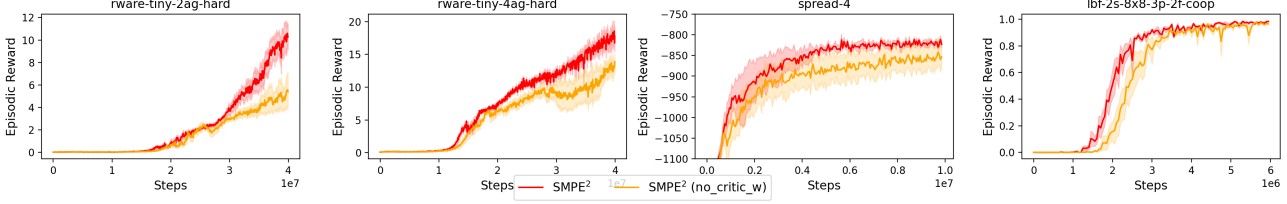

*Figure 18.* Ablation study on learning the state belief representations with respect to policy optimization

### E.4.7. STUDYING THE DYNAMIC NATURE OF THE INTRINSIC REWARD

In this subsection of the Appendix, we examine the dynamic nature of the intrinsic rewards proposed in SMPE². As highlighted in Section 3.2, to enhance stability, we make $z^i$ more stable by avoiding significant changes in the parameters $(\omega_i, \phi_i)$ between successive training episodes. To achieve this, we perform periodic hard updates of $\omega_i$ and $\phi_i$ using a fixed number of training time steps as the update period. Figure 19 shows that the intrinsic rewards do not fluctuate but rather decrease smoothly throughout training until they reach a minimal plateau. Such a behavior is what we expected: the agents to explore the state space and gradually to converge to more deterministic optimal policies.

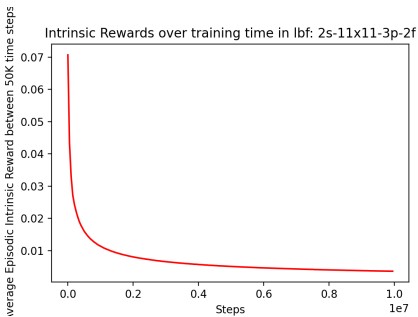

*Figure 19.* Ablation study: Studying the dynamic nature of the intrinsic reward

| Embeddings | Accuracy |
|---|---|
| $z^i$ | **57.5%** |
| $z^i$ (without KL) | 99.3% |
| $z^i$ (without $w^i$) | 80.8% |

*Table 1.* Quantitative ablation study on the embeddings $z^i$ in an LBF task at the 45th time step: Logistic Regression on 5-fold cross validation using the t-SNE representations as the training data and the agent IDs as the labels.

E.4.8. FURTHER T-SNE VISUALIZATIONS AND ANALYSIS OF THE STATE BELIEF EMBEDDINGS

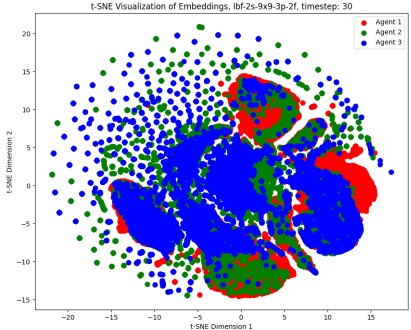

*Figure 20.* SMPE$^2$

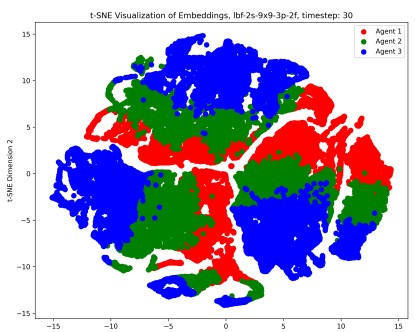

*Figure 21.* SMPE$^2$ ($L_{\text{KL}} = 0$)

We trained a logistic regression classifier using t-SNE representations as training data and agent IDs as labels, and performed standard 5-fold cross validation. Table 1 validates that SMPE$^2$'s beliefs are cohesive, as they are well entangled and difficult to separate (57.5% accuracy). In contrast, SMPE$^2$ without either $L_{\text{KL}}$ or $w^i$ forms much less cohesive beliefs, as it achieves perfect accuracy (99.3%) or high accuracy (80.3%), respectively. We note that the perfect score of SMPE$^2$ without $L_{\text{KL}}$ is due to the fact that if the agents' beliefs do not share the same prior, then they can be formed in totally different (and easily separated) regions. The above results also validate the importance of $w^i$ for producing more cohesive $z^i$.

E.4.9. IS SMPE$^2$ FLEXIBLE? CAN WE USE MAPPO AS THE BACKBONE ALGORITHM?

To address question **Q6**, we evaluate SMPE$^2$ using MAPPO as the backbone algorithm. We denote this method by SMPE_PPO. Figure 22 highlights the flexibility of our method, as SMPE_PPO significantly outperforms its backbone counterpart in all benchmarks. Remarkably, SMPE_PPO almost perfectly solves the *9x9-3p-2f* task, with the backbone MAPPO demonstrating very poor performance.

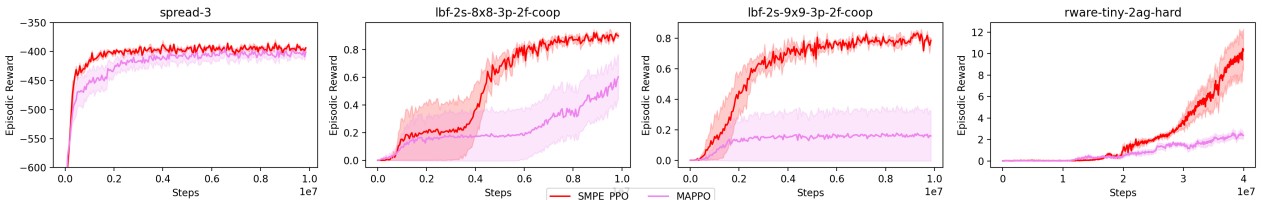

*Figure 22.* SMPE$^2$ is flexible: SMPE_PPO against its backbone algorithm MAPPO

### E.4.10. FURTHER ABLATION STUDY ON THE EXPRESSIVENESS OF $z^i$: LOSS CURVES

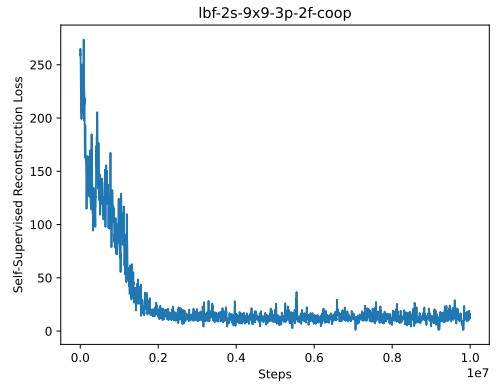

*Figure 23.* Self-supervised reconstruction loss on LBF

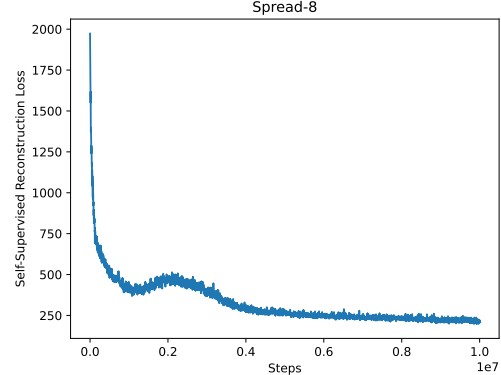

*Figure 24.* Self-supervised reconstruction loss on Spread

Figures 23 and 24 show that the reconstruction loss is decreasing throughout the training of SMPE$^2$, and thus $z^i$, conditioned on $w^i$, is able to reconstruct informative features (identified by $w^i$) of the global state.

### E.4.11. Extra Ablation study on the expressiveness AM filters $w^i$

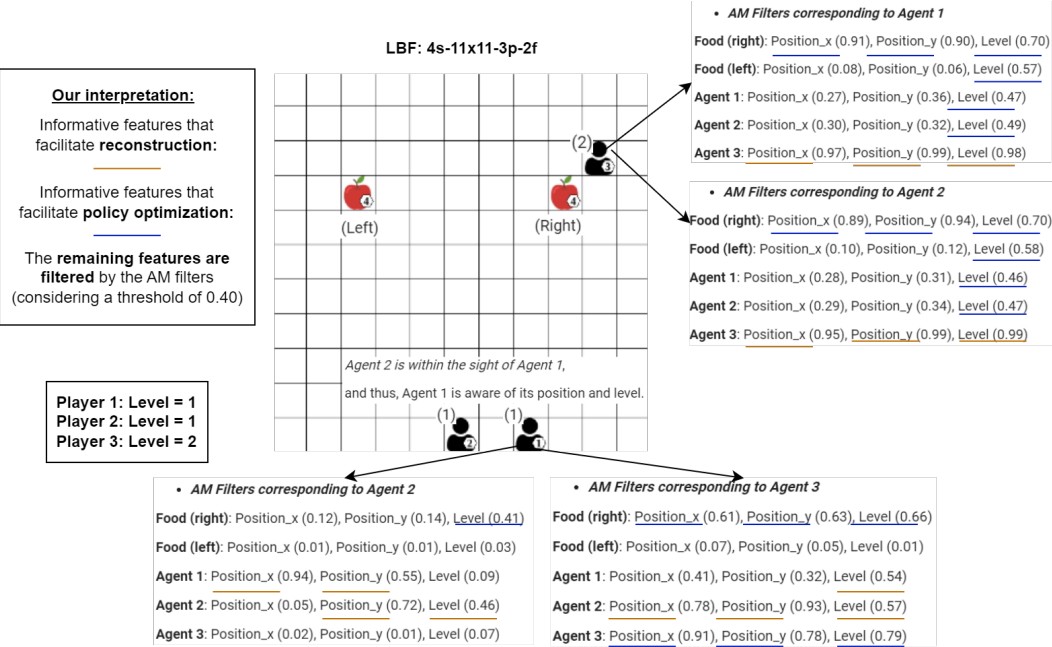

*Figure 25.* Ablation study: Goal: The optimal policy (found by SMPE$^2$) is all players to eat the right food. Agent1: Does not know where both the right food and agent3 are located, or their levels, and seeks to find them. Moreover, it seeks to find if Agent2 sees the right food. Agent2: Has filters very similar to Agent1 (as expected). Agent3: Knows where the right food is located and its level, and seeks to find if the others know it as well and have the required levels for it, as well as if the others are near to the left food. It has very similar filters for both Agent1 and Agent2 (as expected).

In the cooperative LBF task of Figure 25, all players must eat at the same time a food, whose level equals the sum of the players' levels. Each player has a limited sight range of 4 boxes.

Figure 25 demonstrates a global state of the LBF task, along with the AM filters $w^i$ of agents 1 (bottom) and 3 (top). In this state, agent1 and agent2 observe each other (i.e., their locations and levels), while agent3 observes food (right) (i.e., its location and level). The figure is an instance of the best SMPE$^2$ policy, which guides all players to eat food (right) at the same time. In the figure, the features that are not highlighted (with blue/orange) are filtered out as uninformative.

The figure illustrates the expressivity of $w^i$. It shows that uninformative state features are blocked by $w^i$ by assigning them low weights, if either: (a) are easily inferred through reconstruction but are redundant for enriching $z^i$ (e.g., Agent2/Position_x (0.05) see the bottom left filters), or (b) do not help policy learning (e.g., Food(right)/Position_x (0.12) see the bottom left filters). More specifically:

- Regarding Agent1, the agent does not know where Agent3 and food (right) are located, or their levels, and seeks to find them (see the values highlighted with blue). In doing so, the AM filters of Agent1 corresponding to both Agent2 and Agent3 block information about food (left) since it is irrelevant to its policy. Also, the filters corresponding to Agent2 block information regarding both Agent3 and the location of food (right) because they are not observed by Agent2. Moreover, these filters block Agent2/Position_x but not Agent2/Position_y, because Agent1 and Agent2 have the same x-coordinate. We dismissed the AM filters of Agent2, as they were pretty much the same as those of Agent1.

- Regarding Agent3, the agent seeks to find if the others know as well where food (right) is located, and if they have the required levels for it, as well as if the others observe food (left) (see the values highlighted with blue). Its AM filters block (a) the features that describe the location of food (left), and (b) the positions of the other agents. As for the latter, Agent3 is already near food (right), which is the target of the joint policy, and waits for the other agents to appear on its sight in order to eat food (right) at the same time. Moreover, the filters corresponding to both Agent1 and Agent2 are very similar, as expected, because both agents have the same levels and are located next to each other.

- We highlight with orange the high-value features which intuitively were selected by SMPE to facilitate the reconstruction training, helping $z^i$ incorporate useful information about the neighborhood of the agent.

### E.4.12. COMPARISON OF RUNNING TIMES OF THE EXAMINED METHODS

In Table 2, we compare the running times of the examined methods in LBF: *2s-11x11-2p-2f*, using the following specs: a GPU RTX 3080ti, a CPU 11th Gen Intel(R) Core(TM) i9-11900 and a 64 GB RAM.

*Remark* E.2. In our experiments, SMPE[2] is faster approximately 25 times than MASER, 30 times than EMC, 17 times than EOI and 2 times than ATM in lbf:*2s-12x12-2p-2f*.

| MARL Method | Running Time |
|:---:|:---:|
| MAA2C | 0d 00h 37m |
| COMA | 0d 00h 54m |
| ATM | 0d 02h 32m |
| EOI | 0d 17h 11m |
| EMC | 1d 05h 34m |
| MASER | 1d 01h 08m |
| *SMPE*$^2$ | 0d 01h 13m |

*Table 2.* Comparison of running times of the examined methods on lbf:*2s-12x12-2p-2f*

# F. Implementation Details

Our implementation[2] is built on top of the EPyMARL[3] (under the MIT License) python library. For all baselines, we used the open source code of the original paper. SMPE[2] uses the standard MAA2C architecture (see EPyMARL) and three-layer MLPs for both the Encoder-Decoder (ED) and for the AM filters. It is worth noting that as for ED in our experiments we did not observe any further improvement in performance when using standard RNNs (such as GRUs, or LSTM) instead of MLPs. In practice, we update ED parameters every $N_{ED}$ steps. Last but not least, for all benchmark settings, agents (for all algorithms) used shared policy parameters, except for Spread 3-4-5 where each agent had its own parameter weights.

| Name | Description | Value |
|---|---|---|
| $H$ (mpe-lbf-rware) | time horizon | 10M-10M-40M |
| $N_{envs}$ | number of parallel envs of MAA2C | 10 |
| $N_{ep-len}$ (mpe-lbf-rware) | maximum length of an episode | 25-50-500 |
| $N_{\text{tup}}$ | number of environment steps before optimization | 100 |
| $N_D$ | replay buffer capacity (for ED training) | 50,000 |
| $N_{bs}$ | ED replay buffer batch size | 16 |
| $N_{ED}$ | update ED parameters every $N_{ED}$ time steps | 2000 |
| $N_{wtup}$ | update target filter parameters every $N_{wtup}$ time steps | 100,000 |
| $\text{lr}_\pi$ | learning rate for RL algorithm | 0.0005 |
| $\text{lr}_w$ (rware) | learning rate for filter update | 0.0005 (0.00005) |
| $\text{lr}_{w\_critic}$ (mpe, rware) | learning rate for filter update in critic loss | 0.00005 (0.0000005) |
| $\text{lr}_{ED}$ | learning rate for ED | 0.0005 |
| hidden_dim (rware) | hidden dimensionality of Actor and Critic NN | 128 (64) |
| latent_dim (mpe) | latent dimensionality of ED | 32 (64) |
| $\gamma$ | discount factor | 0.99 |
| $\lambda_{\text{rec}}$ (lbf) | lambda coefficient for $L_{\text{rec}}$ | 1 (0.5) |
| $\lambda_{\text{KL}}$ | lambda coefficient for $L_{\text{KL}}$ | 0.1 |
| $\lambda_{\text{norm}}$ (mpe-lbf-rware) | lambda coefficient for $L_{\text{norm}}$ | 0.1-1-0.1 |
| $\beta_H$ | entropy coefficient in policy gradient | 0.01 |
| $\beta$ (lbf-rware) | intrinsic reward coefficient | 0.1-0.001 |
| $shared$ (mpe-spr8-lbf-rware) | shared policy params | F-T-T-T |
| $shared\_ED$ | shared ED params | F (False) |

Table 3. Implementation Details of SMPE[2]

[2]Our official source code can be found at https://github.com/ddaedalus/smpe.
[3]see https://github.com/uoe-agents/epymarl

| Name | Value |
|---|---|
| batch size | 10 |
| hidden dimension | 64 |
| learning rate | 0.0005 |
| reward standardisation | True |
| network type | GRU |
| entropy coefficient | 0.01 |
| target update | 200 |
| buffer size | 10 |
| $\gamma$ | 0.99 |
| observation agent id | True |
| observation last action | True |
| n-step | 5 |
| epochs | 4 |
| clip | 0.2 |

*Table 4.* Hyperparameters for MAPPO

| Name | Value |
|---|---|
| batch size | 32 |
| hidden dimension | 64 |
| learning rate | 0.0005 |
| reward standardisation | True |
| network type | GRU |
| evaluation epsilon | 0.0 |
| epsilon anneal | 50000 |
| epsilon start | 1.0 |
| epsilon finish | 0.05 |
| target update | 200 |
| buffer size | 5000 |
| $\gamma$ | 0.99 |
| observation agent id | True |
| observation last action | True |
| episodic memory capacity | 1000000 |
| episodic latent dimension | 4 |
| soft update weight | 0.005 |
| weighting term $\lambda$ of episodic loss | 0.1 |
| curiosity decay rate ($\eta_t$) | 0.9 |
| number of attention heads | 4 |
| attention regulation coefficient | 0.001 |
| mixing network hidden dimension | 32 |
| hypernetwork dimension | 64 |
| hypernetwork number of layers | 2 |

*Table 5.* Hyperparameters for EMC

| Name | Value |
| --- | --- |
| optimizer | RMSProp |
| hidden dimension | 64 |
| learning rate | 0.0005 |
| reward standardisation | False |
| network type | GRU |
| evaluation epsilon | 0.0 |
| epsilon anneal | 50000 |
| epsilon start | 1.0 |
| epsilon finish | 0.05 |
| target update | 200 |
| buffer size | 5000 |
| $\gamma$ | 0.99 |
| observation agent id | True |
| observation last action | True |
| mixing network hidden dimension | 32 |
| representation network dimension | 128 |
| $\alpha$ | 0.5 |
| $\lambda$ | 0.03 |
| $\lambda_I$ | 0.0008 |
| $\lambda_E$ | 0.00006 |
| $\lambda_D$ | 0.00014 |

*Table 6.* Hyperparameters for MASER

| Name | Value |
| --- | --- |
| optimizer | Adam |
| batch size | 10 |
| hidden dimension | 128 |
| learning rate | 0.0005 |
| reward standardisation | True |
| network type | GRU |
| target update | 200 |
| buffer size | 10 |
| $\gamma$ | 0.99 |
| observation agent id | True |
| observation last action | True |
| n-step | 5 |
| entropy coefficient | 0.01 |
| classifier ($\phi$) learning | 0.0001 |
| classifier ($\phi$) batch size | 256 |
| classifier ($\beta_2$) | 0.1 |

*Table 7.* Hyperparameters for EOI

---

**Algorithm 1** State Modelling for Policy Enhancement and Exploration (SMPE$^2$)

---

1: Initialize $N_{envs}$ parallel environments (required for the backbone MAA2C)
2: Initialize $N$ actor networks with random parameters $\psi_1, \ldots, \psi_N$
3: Initialize the critic networks and their target networks with random parameters $k, \xi$ and $k', \xi'$
4: Initialize $N$ encoder-decoder networks with random parameters $\phi_1, \ldots, \phi_N$ and $\omega_1, \ldots, \omega_N$
5: Initialize a replay buffer $D$ of maximum capacity $N_D$
6: Initialize $N$ weight networks with random parameters $\phi_1^w, \ldots, \phi_N^w$
7: Initialize $N$ target weight networks (for $\tilde{w}^i$) with random parameters $(\phi_1^w)', \ldots, (\phi_N^w)'$
8: **for** time step $t = 1, \ldots, H$ **do**
9:     **for** agent $i = 1, \ldots, N$ **do**
10:        Receive current observation $o_t^i$
11:        Sample belief $z_t^i \sim q_{\phi_i}(z^i \mid o_t^i)$
12:        Sample action $a_t^i$ from $\pi_{\psi_i}(a_t^i \mid h_t^i, z_t^i)$
13:     **end for**
14:     Execute actions and receive states $s_{t+1}$, observations $o_{t+1}$ and the shared reward $r_t$
15:     **for** agent $i = 1, \ldots, N$ **do**
16:        Calculate the intrinsic reward $\hat{r}_t^i$ and compute total reward $\tilde{r}_t^i = r_t + \beta \hat{r}_t^i$ using the $SH$ function, and use $\tilde{r}_t^i$ as the reward of agent $i$
17:     **end for**
18:     Store the received (joint) transition $(s_t, a_t, o_t, \tilde{r}_t, s_{t+1}, o_{t+1})$ in $D$
19:     **if** Parallel Episodes Terminate **then**
20:        **for** agent $i = 1, \ldots, N$ **do**
21:           Update $\psi_i$ on the sampled trajectories by minimizing $L_{\text{actor}}$ of the Equation (8)
22:           Update $\xi$ on the sampled trajectories by minimizing $L_{\text{critic}}$ of the Equation (6)
23:           Update $\phi_i^w$ and $k$ on the sampled trajectories by minimizing $L_{\text{critic}}^w$ of the Equation (6)
24:        **end for**
25:        Restart a new episode for each of $N_{envs}$ parallel environments
26:     **end if**
27:     **if** condition for training the encoder-decoders is met ($N_{ED}$) **then**
28:        **for** agent $i = 1, \ldots, N$ **do**
29:           Update $\omega_i$ on $D$ using batch size $N_{bs}$ by minimizing $L_{\text{rec}}$ (Equation (3)) and $L_{\text{KL}}$ (Equation (5))
30:           Update $\phi_i$ on $D$ using batch size $N_{bs}$ by minimizing $L_{\text{rec}}$ (Equation (3))
31:           Update $\phi_i^w$ on $D$ using batch size $N_{bs}$ by minimizing $L_{\text{rec}}$ (Equation (3)) and $L_{\text{norm}}$ (Equation (4))
32:        **end for**
33:     **end if**
34:     **if** condition for updating the target policy networks is met ($N_{tup}$) **then**
35:        Update $\xi' = \xi$ and $k' = k$
36:     **end if**
37:     **if** condition for updating the target filter networks is met ($N_{wtup}$) **then**
38:        **for** agent $i = 1, \ldots, N$ **do**
39:           Update $(\phi_i^w)' = \phi_i^w$
40:        **end for**
41:     **end if**
42: **end for**

---

