# OpenReview forum: "Enhancing Cooperative Multi-Agent Reinforcement Learning with State Modelling and Adversarial Exploration"
_ICML.cc/2025/Conference — ICML 2025 poster_

### Official Review · Reviewer_MTEX · 2025-03-06

**Overall Recommendation:** 2

**Summary:**

The paper proposes a state modelling framework to infer meaningful beliefs about the unobserved state while filtering redundant information. It reconstructs other agents’ observations using an encoder-decoder. To overcome the sparse reward challenge, this paper proposes a adversarial count-based intrinsic exploration method to encourage the agents to visit novel states.

**Claims And Evidence:**

The motivation of the proposed method is not clearly stated. It is hard to find the real contribution and innovation of the paper from the main contributions elaborated in the introduction.

This paper assumes that states contain information redundant for optimizing agents’
individual policies. Why make such an assumption?

What does the “adversarial targets” mean? Do you mean the target entity in the actual environment?

**Essential References Not Discussed:**

N/A

**Experimental Designs Or Analyses:**

This paper conducted evaluations on MPE, LBF, and RWARE. Experiments show that the proposed method outperforms the baselines in terms of learning speed and final performance. The algorithm has obvious advantages in sparse reward tasks.

**Methods And Evaluation Criteria:**

The paper uses an encoder-decoder to model the other agents’ observations based on only the local observation, which seems reasonable. To improve exploration, the paper encourages diversity of the latent variable z by using count-based intrinsic rewards.

This paper designs an adversarial exploration method to “discover novel, high-value states while improving the discriminative abilities of others”. But the count-based intrinsic reward only can drive agents to visit novel states. Such an intrinsic reward could not serve the purpose of encouraging the exploration of high-value states.

**Other Comments Or Suggestions:**

+ The citation format is incorrect.
+ In many figures, the legends obscure the main figures.
+ “Learning wi (and thus zi) w.r.t. policy optimization” Such a statement is difficult to understand.
+ Figure 1 could be presented in a better way. The current figure is difficult to read.

**Other Strengths And Weaknesses:**

Strengths:
+ The experiments are good.
+ The proposed method is novel.

Main weaknesses:
+ The overall presentation of the paper lacks clarity. The writing of this paper needs further improvement. I believe that the quality of the paper will be greatly improved after improving the writing.

**Questions For Authors:**

+ What does the adversarial exploration and discriminative abilities mean? How you improve the discriminative abilities of other agents?
+ The method relies solely on local observations for agent modeling. Does it scale well as the number of agents increases?

**Relation To Broader Scientific Literature:**

This paper is linked to agent modeling and exploration in MARL.

**Theoretical Claims:**

I have checked the proof of the proposition 2.1.

---

> ### Author Rebuttal · Authors · 2025-04-01
>
> We sincerely appreciate your time and input. We respond to your comments and questions below.
>
> > 1. The motivation of the proposed method is not clearly stated.
>
> We respectfully disagree with the reviewer's comment. In Lines 28-36 (right), we clearly state our main motivation for this paper: "we are interested in settings where agents lack explicit communication channels during execution. Such settings are of particular interest, because, while communication-based methods leverage inexpensive simulators for training, they may incur substantial computational overhead when executed in real-world environments". Moreover, in the intro (Lines 43 right– 64 left), we provide a detailed discussion of significant drawbacks and problematic assumptions of existing agent modelling (AM) methods. Remarkably, many of these claims (e.g., redundant state information, AM without exploration and policy optimization, single-agent AM) are empirically validated by our results and the extensive ablation study, demonstrating their detrimental impact on MARL performance. These challenges highlight the need for a more principled approach to AM in MARL and further motivate our approach.
>
> > 2. Real contributions of the paper
>
> Due to space constraints, please see our response to Question 7. in the rebuttal of reviewer CpzR.
>
> > 3. This paper assumes that states contain information redundant for optimizing agents’ individual policies. Why?
>
> This is an established assumption which has been validated by the NeurIPS paper ([48]-th reference in our paper) and the AAMAS paper [1], supported both by the curse of dimensionality and by intuitive reasoning. Specifically, in large and complex state spaces, each agent naturally prioritizes information that is more relevant or proximal to them while assigning lower importance to remote or less goal-related information.
>
> > 4. What does the “adversarial targets” mean? Do you mean the target entity in the actual environment?
>
> The term "targets" has been used many times before the adversarial exploration section (e.g., see lines 155 right, 214 left, 175 right, 181 right, 185 right, 187 right, 195 right) in the paper to denote the reconstruction targets of the encoder-decoder.
>
> > 5. This paper designs an adversarial exploration method to “discover novel, high-value states while improving the discriminative abilities of others”. But the count-based intrinsic reward only can drive agents to visit novel states. Such an intrinsic reward could not serve the purpose of encouraging the exploration of high-value states.
>
> Thanks for this comment. By the means of adversarial exploration, each agent is  motivated to reach novel states, while she challenges the state modelling abilities of other agents. By doing so, our exploration schema aims at improving the joint belief about the joint state, and thus the joint policy, as the policy network of each agent uses her belief as extra input. Therefore, our method is more likely to reach novel, high-value states than the original which hashes observations.
>
> > 6. The citation format is incorrect. In many figures, the legends obscure the main figures. Figure 1 could be presented in a better way.
>
> Thanks for your comments. We will fix these minor typos in the camera-ready version. Regarding Fig. 1, we have thoroughly explain its components in high-level in Lines 147 (right) - 169 (left).
>
> > 7. What do the adversarial exploration and discriminative abilities mean?
>
> Thanks for this question. By discriminative abilities of an agent under partial observability we mean the ability of the agent to identify meaningful information about the unobserved global state based on her local information. In other words, discriminative abilities represent how well the agent models the global state. Our approach defines the meaningful state information within the state modelling framework (see lines 129 left - 158 left).
>
> Adversarial exploration is extensively explained and motivated in Lines 246 (left) - 267 (left).
>
> > 8. The overall presentation of the paper lacks clarity
>
> We kindly disagree with the above comment that there are major presentation issues about the paper. Reviewer hakk finds that "*the paper was clear and easy to follow, which made understanding the ideas very smooth and engaging*". Some of the points you found unclear are thoroughly explained earlier in the paper, so we believe there is a chance that you missed these explanations and this rendered the points, our motivation and contributions unclear. If this is the case, we hope through a more careful reading to appreciate our ideas and results more. Otherwise, if you insist that there are specific points which require better explanation from our end, please let us know to fix them.
>
> > 9. Scalability with more agents
>
> Thanks for this question. See our response to reviewer hakk on question 1.
>
> [1] Li et al. From explicit communication to tacit cooperation: A novel paradigm for cooperative MARL. 2024

---

### Official Review · Reviewer_NMyw · 2025-03-11

**Overall Recommendation:** 1

**Summary:**

The paper presents a novel approach to cooperative multi-agent reinforcement learning (MARL) under partial observability by introducing a state modelling framework combined with adversarial exploration. In this framework, each agent infers a latent belief from its local observation using a variational encoder–decoder, while learnable agent modeling filters remove redundant features to capture essential global state information. The resulting SMPE2 algorithm leverages these latent beliefs by incorporating them into the policy network and using count-based intrinsic rewards to encourage exploration of novel, high-value states, thereby enhancing coordination and overcoming sparse-reward challenges. Experimental results on benchmarks like MPE, LBF, and RWARE show that SMPE2 outperforms state-of-the-art methods, leading to faster convergence and higher episodic rewards.

## Update After Rebuttal
Thanks for the authors' rebuttal. However, I still feel like this paper is short of reasonable justifications. The explanations provided by the authors stand on their own suppose with no sufficient evidence to back up. Although the authors emphasized their work is theoretically sound, I have different understanding on this point. As a result, I think of this paper is not ready for publication at this moment. I suggest the authors may link their modeling and approach to some realistic implication, which is always the aim of engineering.

**Claims And Evidence:**

### Supported Claims
1. The authors have empirically demonstrated the effectiveness of each component of the proposed algorithm SMPE2.
2. The authors claimed that "The framework assumes that the joint state information can be redundant and needs to be appropriately filtered in order to be informative to agents." This has been partially verified by visualizing weight functions.
3. The authors claimed that "Intuitively, $w_{i}$ has an AM interpretation, as it represents the importance of each of other agents’ information to agent i’s state modelling." This has been verified by one example.
4. The authors claimed that "Note the importance of the AM filter $w_{i}$: (a) With it, although the target of ED grows linearly
with the number of agents, only features that can be inferred through $z_{i}$ remain as part of other agents’ observations in the reconstruction loss. (b) Without it, it would be challenging to infer meaningful embeddings $z_{i}$, due to non-informative joint state information." These two conditions have been verified by showing ablation study and visualizing the the projected $z_{i}$ by t-SNE.

### Problematic Claims
1. In introduction (Line 59-64), the authors blame that previous work drew multiple assumptions. This is not convincing to be a motivation. The main reason is that those assumptions are primarily used to estimate the boundary of methods. In other words, no assumptions does not imply the effectiveness to all scenarios, except that you can prove that rigorously in mathematics or do experimentation in all possible scenarios. Especially, I find that this paper also made several assumptions, such as the one illustrates the relation between latent space and observation (Line 181-184).
2. The authors claimed that the belief $z_{i}$ contains meaningful information about the unobserved state, informative for optimizing the agent's own policy. This assumption is quite strong, for example, how is this guaranteed?
3. As for the definition of conditions for non-informative joint state feature, the authors claimed that "it cannot be inferred through $z_{i}$, in the sense that the agent cannot predict it conditioned on its own observation due to partial observability and non-stationarity." This condition is extremely strong, as it rules out information that may be beneficial to agent decision making, but not acquired due to the belief space the authors defined. As I haven't seen any clear and concrete definition for the belief space, this condition is vague. I suggest the authors can have more rigorous discussion here.
4. The authors claimed that "using the full state information as an extra input to the policy, even when utilizing a compressed embedding, may harm performance due to redundant state information non-informative to agent $i$." This may be due to the information loss brought by compression to get $z_{i}$, rather than the redundant state information.
5. The authors claimed that "Following the definition of the state modelling problem, we aim to ensure that $w_{i}$ (and thus $z_{i}$) incorporate information relevant to maximizing $V^{\pi}$ and thus, $w_{i}$ to be capable of filtering non-informative state features irrelevant to maximizing agent’s future rewards." Although the weight $w_{i}$ is learnable, I am afraid it could be not effective to filter out the non-informative information. For example, even though the weight converges to including partial non-informative, I believe it can still find a local optimum by policy. More importantly, this suboptimality is difficult to be verified.

**Essential References Not Discussed:**

No.

**Experimental Designs Or Analyses:**

This paper has designed plenty of experiments, motivated by several research questions. I have checked these in detail, and I believe all these are sound. In addition, the experimental analyses also seem reasonable.

**Methods And Evaluation Criteria:**

### Methods
From the high-level view, the proposed method makes sense to the research problem this paper aims to solve. However, as the proposed method is constituted of multiple components, it is difficult to make a coherent understanding of the proposed method from the theoretical perspective.

### Evaluation Criteria
This paper is different from other regular papers in MARL. It has conducted a lot of abalation studies from diverse dimensions, such as visualizing the learned features. This is the most critical benefit of this paper.

**Other Comments Or Suggestions:**

I suggest the authors can re-organize the writing in a more logical way, to emphasize the most critical contribution delivered by this paper.

**Other Strengths And Weaknesses:**

### Other Weaknesses
This paper is not easy to comprehend. The main reason is that it has included too much information, with an overall description that establish a logic chain through all these components of the proposed method. Moreover, it is not easy to track the main contribution of this paper. It seems like a compound of several technical tricks.

**Questions For Authors:**

1. The authors are requested to address the concerns in **Claims And Evidence**.

2. In addition, I believe MAVEN [1] is highly related to the framework proposed in this paper, except that it does not include an additional intrinsic reward term. For this reason, I would like to see the additional experiments for comparison between the proposed method and MAVEN.

3. The authors claimed that "Agent is intrinsically motivated to discover novel $o_{i}$ (which lead to novel $z_{i}$) which at the same time constitute unseen targets for the others’ reconstruction training. Therefore, these targets aim to adversarially increase the losses of other agents’ reconstruction models." Do the authors mean the mismatch between each agent's information due to independent update?

4. The authors claimed that "To do so, given that $z_{i}$ is solely conditioned on $o_{i}$, the agent is implicitly motivated to discover novel observations that must lead to novel $z_{i}$." What is the logic here? Can the authors provide more evidence to clarify this claim?


[1] Mahajan, Anuj, et al. "Maven: Multi-agent variational exploration." Advances in neural information processing systems 32 (2019).

**Relation To Broader Scientific Literature:**

This paper mainly investigates a long-standing problem in collaborative MARL, exploration and improving coordination in decentralized policies (or independent learning). The general framework proposed in this paper has no big difference from the previous work, for example, learning embeddings representing other agent behaviors [1,2] and using intrinsic reward to improve exploration [3].


[1] Papoudakis, Georgios, Filippos Christianos, and Stefano Albrecht. "Agent modelling under partial observability for deep reinforcement learning." Advances in Neural Information Processing Systems 34 (2021): 19210-19222.

[2] Mahajan, Anuj, et al. "Maven: Multi-agent variational exploration." Advances in neural information processing systems 32 (2019).

[3] Pathak, Deepak, et al. "Curiosity-driven exploration by self-supervised prediction." International conference on machine learning. PMLR, 2017.

**Theoretical Claims:**

This paper has a simple theoretical claim. I have checked the proof and it should be correct.

---

> ### Author Rebuttal · Authors · 2025-04-01
>
> > 1. Motivation and Discussion about Assumptions of related work
>
> We respectfully disagree with the reviewer's comment. In the introduction (Lines 43–64), we provide a detailed discussion of significant drawbacks and problematic assumptions of existing agent modelling (AM) methods. Remarkably, many of these claims (e.g., redundant state information, AM without exploration and policy optimization, single-agent AM) are empirically validated by our results and the extensive ablation study, demonstrating their detrimental impact on MARL performance. These challenges highlight the need for a more principled approach to AM in MARL. Regarding Lines 59–64, we believe that the assumptions outlined — namely, a priori knowledge of state features, centralized execution, and a focus solely on team games — are quite restrictive. These constraints limit the practicality of such algorithms, making them less applicable to a wide range of real-world scenarios.
>
> > 2. Assumption: z_i contains meaningful information about the unobserved state, informative for optimizing the policy.
>
> Assuming that there exists a latent belief space which contains unobserved state information meaningful for optimizing the policy has been very standard in AM (e.g., see [40, 42, 44]). We note that this  is the main assumption in order for AM to be applicable under partial observability.
>
> > 3. It rules out information that may be beneficial to decision making, but not acquired due to the defined beliefs.
>
> The reviewer must have misunderstood that according to our definition if a state feature is beneficial to decision making, then the framework considers this feature to be "informative" (see first bullet, line 139). Such a feature could be identified by w_i, and thus z_i, being conditioned on w_i, is informative of this feature during exploration. Also, our definition implies that if a feature cannot be inferred through z_i and it is not relevant to policy optimization, then this feature is "not-informative".
>
> > 4. Comment on "even when utilizing a compressed embedding"
>
> Thanks for this comment. It is known from [9] that providing full state information as an additional input to the policy can degrade performance due to redundant, non-informative state features. In our work, we demonstrate that this issue can persist even when using a compressed embedding z_i, as shown in Fig. 5 (left), where applying filters leads to improved performance. We will ensure that this detail is clarified in the final version.
>
> > 5. Even though the weight converges to including partial non-informative, it can still find a local optimum by policy. More importantly, this suboptimality is difficult to be verified.
>
> Due to space constraints, please see our response to Question 8. in the rebuttal of reviewer CpzR.
>
> > 6. Difference from LIAM, MAVEN and [3]
>
> Due to space constraints, please see our response to Question 10. in the rebuttal of reviewer W3Cu.
>
> > 7. Multiple components: difficult to understand from the theoretical perspective
>
> We believe that all loss components have been well-explained in Sec. 3 and are well-validated by a plethora of experiments (see Fig. 5, 7, 15 - 26). More specifically, from a theoretical standpoint we give detailed intuition of each component:
> 1. L_rec: 172 left - 182 right
> 2. L_wcritic: 213 left - 218 left
> 3. intrinsic reward: Sec. 3.2
> 4. L_norm: 196 left - 200 left
> 5. L_KL: 202 left - 207 left
>
> Our method does not rely on "tricks" but on **conceptual components** (1–3), motivated by how we think of the ideal MARL behavior against partial observability (see Lines 77-92 (left) and 98-104 (right)). The components 4–5 were introduced to ensure that the proposed method is theoretically sound (also illustrated in Fig. 7, 17) and adheres to our definition of state modeling (lines 152 & 205 left).
>
> > 8. Not easy to track the main contribution of this paper
>
> Due to space constraints, please see our response to Question 7. in the rebuttal of reviewer CpzR.
>
> > 9. Comparison to MAVEN
>
> Due to space constraints, please see our response to Question 6. in the rebuttal of reviewer CpzR.
>
> > 10. Question about "intrinsically motivated...models"
>
> Since each agent updates its policy independently, conditioned only on its own observations and beliefs, it is intrinsically motivated to seek novel observations o_i, thereby discovering novel z_i. However, since o_i serve as part of the target for other agents' decoders, they inadvertently increase the reconstruction loss of those agents.
>
> > 11. Question about "To do so...z_i."
>
> Since z_i is a function of only o_i, and agent i is motivated to reach novel z_i, then the only way for the agent to achieve this is by discovering novel o_i that lead to novel z_i. In Appendix E.4.7 we show how we effectively handle the dynamic nature of the reward.
>
> [@] Papadopoulos et al. (AAMAS 2025) - An Extended Benchmarking of Multi-Agent Reinforcement Learning Algorithms in Complex Fully Cooperative Tasks.

---

### Official Review · Reviewer_hakk · 2025-03-13

**Overall Recommendation:** 2

**Summary:**

In most Multi-Agent Reinforcement Learning (MARL) problems, agents operate under partial observability, making decisions based on their observations and beliefs rather than the full state, and a naïve integration of the full state to each agent’s observation can introduce irrelevant information, hinder exploration, and degrade performance. To address this, the paper proposes a state modelling framework that enables agents to construct meaningful beliefs about the unobserved state, optimizing their policies and improving exploration.  The authors introduce State Modelling for Policy Enhancement through Exploration (SMPE2), which consists of two components: (1) self-supervised state modelling, where an encoder-decoder predicts other agents' observations using only local information while Agent Modelling filters that remove redundant joint-state features, and (2) adversarial count-based intrinsic exploration, an intrinsic reward mechanism that uses SimHash-based novelty detection to guide exploration toward novel, high-value states. Empirical results on MPE, LBF, and RWARE show SMPE2 outperforms state-of-the-art MARL baselines, with extensive ablation studies confirming the importance of its components.

**Claims And Evidence:**

The paper claims that the state modelling objective is equivalent to the Dec-POMDP objective, providing a proof in Appendix D.

Their claim that state modelling improves MARL performance is supported by empirical evidence from a variety of scenarios: dense (MPE), balanced sparse (LBF), and very sparse (RWARE) reward settings. Additionally, they demonstrate that SMPE2 is flexible, as it can be applied to different MARL algorithms, where they experiment in the main paper using MAA2C as the backbone. Additional results in the appendix show SMPE2-MAPPO outperforming MAPPO as well.
The authors further claim that AM filters retain only informative state features, preventing redundant information from degrading performance, while adversarial exploration guides agents toward high-value states, improving cooperation and exploration. Ablation studies confirm that removing these components hinders learning and slows convergence.

A major concern regarding the validity of these claims is the performance of SOTA algorithms on the suggested scenarios. To my knowledge, the only literature that uses RWARE hard scenarios is [1] which compare their work using different baselines, besides a recent paper [2] that shared the same scenarios choice, however, another study using a JAXified version of these environments [3] suggests that MAPPO performs well in the proposed RWARE settings.

**Reference:**

[1] Christianos et al. (2021) - Shared Experience Actor-Critic for Multi-Agent Reinforcement Learning.

[2] Papadopoulos et al. (2025) - An Extended Benchmarking of Multi-Agent Reinforcement Learning Algorithms in Complex Fully Cooperative Tasks.

[3] Mahjoub et al. (2025) - Sable: A Performant, Efficient, and Scalable Sequence Model for MARL.

**Essential References Not Discussed:**

Despite using the transformer-based algorithm ATM, the paper lacks discussion on other SOTA transformer-based MARL methods, such as [7], which explore sequence-based representation learning as an alternative to state modelling. Additionally, it does not reference heterogeneous-agent reinforcement learning, such as [8], which studies how agents with different capabilities and roles coordinate in MARL settings. And lastly, the paper did not mention Shared Experience Actor-Critic (SEAC) [1], which focuses on improving exploration by sharing experiences among agents.

**References:**

[7] Wen et al. (2022) - Multi-Agent Reinforcement Learning as a Sequence Modeling Problem.

 [8] Zhong et al. (2023) - Heterogeneous-Agent Reinforcement Learning

**Experimental Designs Or Analyses:**

The experiments are well-structured, with effective ablation studies highlighting each SMPE2 component’s impact. However, the authos evaluate their method using customized scenarios and ones not commonly used in prior work, including additional well-established tasks from the same settings (MPE, LBF and Rware) and additional environments in general would provide more reliable comparisons and further validate SMPE2. Additionally, I would suggest to briefly explaining t-SNE before using it in Figure 7, as some readers may not be familiar with its purpose in the ablation study.

**Methods And Evaluation Criteria:**

The paper evaluates SMPE2 in three environments  (MPE, LBF, and RWARE) and compares it against MAA2C, COMA, MAPPO, ATM, EMC, MASER, and EOI. The selection of some baselines is well motivated, as some incorporate state modelling per agent, such as EOI, and MASER. The authors provide detailed information on computational resources, hyperparameters, and benchmark settings, ensuring reproducibility.

However, I have a few concerns regarding the evaluation:

1. Scenarios selection: The authors use customized LBF and hard RWARE scenarios to showcase exploration skills. While this approach is valid, it would be beneficial to reinforce these findings on well-established scenarios from the literature [4]. This would allow for more direct comparisons with prior works.
2. Scalability and harder settings: Further evaluation on larger-scale environments, such as RWARE larger grid and more agents (e.g., customized environments like large-8ag), could better assess SMPE2's performance in harder exploration settings with more agents. Given that many SOTA MARL algorithms may struggle to provide any signal in such settings, testing on these variants would further reinforce confidence in SMPE2's state modelling and exploration strategies.
3. Hyperparameter tuning details: The paper does not specify how hyperparameters were tuned, how many trials were conducted, or whether tuning was performed per scenario or per environment.
4. Suggestion to increase the number of seeds: The experiments are conducted on six random seeds, but increasing to ten seeds would improve statistical robustness, as discussed in [5,6].

**References:**

 [4] Papoudakis et al. (2021) - Benchmarking Multi-Agent Deep Reinforcement Learning Algorithms in Cooperative Tasks.

 [5] Agarwal et al. (2021) - Deep RL at the Edge of the Statistical Precipice.

 [6] Gorsane et al. (2022) - Standardized Performance Evaluation in Cooperative MARL.

**Other Comments Or Suggestions:**

The paper was clear and easy to follow, which made understanding the ideas very smooth and engaging.

**Syntax Suggestions:**

1. Line 44:  A comma should be added for better readability.
2. "Due to space constraints, ...": This phrase appears multiple times, but it would be better to directly reference the appendix section without justifying why it's not in the main text.
3. Line 173 ("we aim agents") :  This phrasing feels incorrect; rewording would improve clarity.

**Plot Suggestions:**

1. Legends placement: Currently, all legends are placed on top of the x-axis. It would be better to move them above the plots or slightly below for visibility.
2. Figure 5 (left plot): The legend is cropped at the edge, slightly reducing the plot's clarity.
3. Figure 14: The legend redundantly repeats algorithm names twice.

**Clarity Suggestion:**

- The paper uses the format “(number)” for both citations and equations, which can be confusing for readers when referencing them in text. It would help to distinguish them visually, perhaps by adding brackets or a different formatting style for references.

**Other Strengths And Weaknesses:**

**Strengths:** The work is novel and provides thorough ablation studies that effectively address most questions regarding component choices.

**Weaknesses:** The computational overhead is a concern, especially when scaling to larger agent populations with such a complex network. The authors report that SMPE2 is approximately 25× faster than MASER, 30× faster than EMC, 17× faster than EOI, and 2× faster than ATM, but this comparison was done on LBF:2s-12x12-2p-2f, which only includes two agents. It remains unclear whether this speed advantage holds as the number of agents increases.

**Questions For Authors:**

First of all, I would like to acknowledge the thorough ablation studies in the paper. Each time I had a question about a component choice, I later found that an ablation study had already addressed it, which was great to see.

That said, I still have a few questions:

1. While the results demonstrate SMPE2’s strong performance on the tasks considered, it would be helpful to explain in more detail why certain SOTA algorithms struggle in specific scenarios. Additionally, could you provide more details on the hyperparameter tuning process: how were the parameters selected? Lastly, I strongly encourage testing on well-established scenarios from the literature (e.g., from [4]), as this would allow for direct comparisons with prior works and further reinforce SMPE2’s improvements.
2. The reported speedup comparisons are based on LBF with two agents. Does this efficiency hold for larger-scale environments, or would state modelling and AM filters introduce bottlenecks?
3. Given that ATM (a transformer-based algorithm) was included, why was Multi-Agent Transformer (MAT) [7] not considered? Would SMPE2’s state modelling be complementary or redundant in transformer-based architectures?
4. SMPE2 uses parameter sharing in LBF and RWARE but not in MPE. Did you test fully independent policies across all environments, and how does removing parameter sharing affect performance? Additionally, do the baseline algorithms use parameter sharing?

**Relation To Broader Scientific Literature:**

The paper builds on opponent modelling approaches like LIAM, SIDE, and MAVEN, but focuses on improving state representation learning rather than directly modelling other agents. The SimHash-based intrinsic reward mechanism aligns with previous count-based exploration techniques but introduces an adversarial component, distinguishing it from prior methods. In CTDE-based MARL, SMPE2 follows the centralized training decentralized execution (CTDE) paradigm, similar to MAPPO and COMA, but incorporates self-supervised learning for belief state inference, making it a novel contribution to the field.

**Theoretical Claims:**

The paper presents Proposition 2.1, claiming that the state modelling objective is equivalent to the Dec-POMDP objective.

---

> ### Author Rebuttal · Authors · 2025-03-31
>
> We sincerely appreciate your time and input. Please see our responses below:
>
> > 1. Scalability with more agents
>
> Thanks for this question. Along with Spread-8 and LBF 7s-20x20-5p-3f (see Fig. 2, 3), we also add results on other large LBF tasks: namely 8s-25x25-8p-5f and 7s-30x30-7p-4f. We note that these tasks have been benchmarked by the AAMAS2025 paper [1]. Below, we show that SMPE outperforms all baselines in these tasks as well.
>
> |Env |SMPE |  MAA2C |MAPPO|MASER| EOI | ATM
> |-----------------:|---------:|--------:|----------:|----------:|------:|-----:|
> | 8s-25x25-8p-5f| **0.64 +- 0.12**|0.52 +- 0.24|0.41 +- 0.23|0.01 +- 0| 0.07 +- 0.02 | 0.36 +- 0.28 |
> | 7s-30x30-7p-4f| **0.74 +- 0.02**|0.71 +- 0.02   |0.57 +- 0.03|0.01 +- 0| 0.04 +- 0.01 | 0.59 +- 0.04 |
>
>
> > 2. Computational overhead: SM and AM filters a bottleneck?
>
> Thanks for this question. From the table below, we observe that even in larger settings, SM and AM filters only introduce a moderate (and completely justified) extra computational overhead, less than ATM, and far less than EOI, EMC and MASER.
>
> |Env|SMPE |MAA2C|MAPPO|MASER |EOI|ATM|EMC|
> |-----------------:|---------:|--------:|----------:|----------:|------:|-----:|-----:|
> | 25x25-8p-5f| 0d 7h |0d 2h | 0d 3h | 1d 5h| 1d 20h  | 0d 8h| 4d 12h|
> | Spread-8| 0d 9h |0d 4h|0d 5h |2d 5h |3d 9h | 0d 18h | 5d 3h|
>
> > 3. Evaluated scenarios
>
> Thanks for this comment. All LBF and RWARE tasks from [41] are not difficult to solve. More specifically, the RWARE tasks from [41] are not the -hard ones. Moreover, in all LBF tasks from [41] (most of them are cooperative-competitive, that is an easier version of the problem), our method converges fast to an optimal policy. This is the main reason we selected more challenging scenarios from these benchmarks.
>
> Remarkably, [1] benchmarks all of our selected tasks and explicitly highlights as open challenges most of our LBF tasks along with MPE tasks with 5 or more agents, in all of which SMPE performs significantly better than all baselines.
>
> > 4. Why was MAT not considered? SM in transformer-based architectures?
>
> Thanks for this question. Since both ATM and MAT are transformer-based algorithms, we chose to include only one of them to ensure a more diverse set of baselines, rather than over-representing algorithms from the same direction of approaches. We selected ATM over MAT because MAT is not implemented in (E)PyMARL, whereas ATM and all other baselines we consider are. This ensures that all methods are evaluated under the same protocol. Additionally, MAT has already been benchmarked in the same environments (see [1]), where it generally underperforms compared to our baselines. Notably, the authors have not highlighted MAT as a "best" algorithm. Regarding if SMPE would fit a transformer-based architecture, we have not evaluated it yet and we have left this direction as a future work.
>
> > 5. hyperparameter tuning, such as the number of trials, tuning methodology, whether tuning was done per scenario or per environment, and parameter sharing.
>
> Thanks for this comment. We followed [41]: For hyperparameters, optimization was performed for each algorithm separately in each environment (not scenario). Then from each environment, we selected one task and optimised the hyperparameters of all algorithms in this task. We evaluated algorithms using 6 independent runs (seeds).
>
> Except for MPE, where [41] (and also [1]) noted that parameter sharing was more detrimental, we used parameter sharing for all other tasks across all algorithms. All evaluated algorithms ran with the same configuration in parameter sharing. We selected the configuration of parameter sharing based on [1,41], and we did not run further experiments with independent/sharing policies.
>
> > 6. Validity of MAPPO results
>
> Thanks for this comment. We evaluated MAPPO using the EPyMARL code and adhered to the same parameters as in [41]. We are confident that all baselines were run correctly, and we can upload all log files to a GitHub repository after the camera-ready version. As the reviewer pointed out, our MAPPO results are on par with those reported in the AAMAS 2025 paper [1], where we verified that the algorithm and environment hyperparameters were the same. In contrast, [2] does not use the (E)PyMARL library, which could lead to significant differences in the evaluation protocol. Moreover, we find it peculiar that MAPPO manages to reach good performance far before 20M steps (e.g. in 2ag-tiny-hard, 4ag-small-hard). We also conducted additional experiments with different random seeds, but we did not observe the behavior described.
>
> > 7. References Not Discussed and further suggestions
>
> Thanks for this comment. We will make sure to incorporate all suggestions.
>
> [1] Papadopoulos et al. (2025) - An Extended Benchmarking of Multi-Agent Reinforcement Learning Algorithms in Complex Fully Cooperative Tasks.
>
> [2] Mahjoub et al. (2025) - Sable: A Performant, Efficient, and Scalable Sequence Model for MARL.

---

### Official Review · Reviewer_CpzR · 2025-03-17

**Overall Recommendation:** 4

**Summary:**

This paper proposes State Modelling for Policy Enhancement through Exploration, a novel approach to cooperative multi-agent reinforcement learning in partially observable environments without communication. The method enables agents to infer meaningful belief representations about unobservable states through variational inference and self-supervised learning, while filtering out redundant information. The authors claim to enhance agents' policies both explicitly by incorporating these beliefs into policy networks and implicitly through adversarial exploration. Experiments across three benchmarks demonstrate that the proposed consistently outperforms state-of-the-art MARL algorithms, particularly in cooperative tasks that require extensive coordination and exploration.

**Claims And Evidence:**

The authors provide theoretical justification (Proposition 2.1) showing that their state modelling objective equals the DecPOMDP objective. They perform extensive experiments on three benchmark environments (MPE, LBF, and RWARE) against multiple baselines with results that agree with their claims.

**Essential References Not Discussed:**

The paper appears to cover the most relevant related works in MARL, agent modelling, and exploration. However, it could benefit from discussing:
1) More recent work on belief representation learning in MARL, such as approaches using transformers or other sequence modeling techniques to handle partial observability.
2) More extensive comparison with world models or model-based MARL approaches

**Experimental Designs Or Analyses:**

The experimental designs and analyses appear sound. The authors use established benchmarks with appropriate configurations and metrics. The ablation studies are well-designed to isolate the contributions of different components.

One concern I have is that some figures show inconclusive results. A more detailed analysis in the style of rliable [1] could be more informative than some subfigures in Fig 3 and 4.
One minor concern is that the evaluation in MPE Spread with increasing number of agents (3, 4, 5, 8) doesn't fully demonstrate the scalability of the approach with even larger numbers of agents.

[1] Agarwar et al, Deep Reinforcement Learning at the Edge of the Statistical Precipice

**Methods And Evaluation Criteria:**

The evaluation metric (unnormalized average episodic reward) is common. The authors report results with confidence intervals averaged over six random seeds, which is sufficient for statistical significance. I appreciate not doing mean and standard error. The authors also properly analyze their method through ablation studies to verify the contribution of each component.

**Other Comments Or Suggestions:**

Check Figure 1 and more scalability analysis. Figure 3 and 4 have inconclusive results in some subfigures, it is very clear which ones.

Line 262 first column has a typo: “By doing do”

The Guan et al. paper is mentioned twice in the reference section. Both as 9 and 10. Please fix

**Other Strengths And Weaknesses:**

The paper is well-written with consistent notations and the clearly stated questions throughout the main body make the paper easy to follow. The formulas are clear and the notation is consistent. Figure 1 is not entirely clear and the arrow convention is not sufficient to distinguish the gradient and the data flow. This work could benefit a lot from more justification on how the baselines are selected, especially the ones that are fairly simpler than the proposed method.

The paper has a clear ablation study but I would like to see more details on how a method with this many components can scale up to more complex settings.

**Questions For Authors:**

NA

**Relation To Broader Scientific Literature:**

The state modelling framework extends previous agent modelling approaches by learning representations with respect to policy optimization rather than as an auxiliary task. This addresses a limitation in previous works such as "Agent modelling under partial observability for deep reinforcement learning" (Papoudakis et al., 2021) where the aim is to learn a relationship between the trajectory of the controlled agent and the modelled agent. The paper also suggests the use of adversarial methods.
The use of AM filters to handle redundant state information is motivated by findings in "Efficient multi-agent communication via self-supervised information aggregation" (Guan et al., 2022) where the authors aggregate the information through an attention mechanism. Note that the authors mention the same work twice for no reason as two separate entries.
The adversarial exploration schema uses hashing as done in "# exploration: A study of count-based exploration for deep reinforcement learning" (Tang et al., 2017), but applies it in a novel way to the multi-agent setting. I really like the paper’s study of the smoothness of the intrinsic reward smoothness rate.

**Theoretical Claims:**

The authors refer the proof to Proposition 2.1 as “missing proof”. I am not sure what that entails. The proof is sound and follows from the fact that the set of policies encompassed by the state modelling framework includes all policies that could solve the original problem. However, I am not familiar with this methodology.

---

> ### Author Rebuttal · Authors · 2025-04-01
>
> We sincerely appreciate your time and input, along with the positive evaluation. We respond to your comments and questions below.
>
> > 1. A more detailed analysis in the style of rliable [42] could be more informative than some subfigures in Fig 3 and 4.
>
> Thanks for this comment. We will consider using rliable in the camere-ready version.
>
> > 2. Scalability with more agents
>
> See our response in question 2. to reviewer W3Cu
>
> > 3. Comment on missing related works
>
> Thank you for this comment. Does the reviewer some specific reference in mind? We will be happy to add missing related works.
>
> > 4. Figure 1 is not entirely clear and the arrow convention is not sufficient to distinguish the gradient and the data flow.
>
> Thank you for this comment. Our overview of SMPE^2 and the arrow convention follows a visualization approach similar to that of many well-established MARL papers (see [43], [44], [46]). If the reviewer still has objections about the figure, we are open to discuss possible improvements.
>
> > 5. Note that the authors mention the same work twice for no reason as two separate entries. Line 262 first column has a typo: “By doing do”. The Guan et al. paper is mentioned twice in the references. Line 1194 should be well-entagled instead
>
> Thanks! We will fix these minor typos.
>
> **Due to space constraints, below we include responses to some of the questions of the other reviewers.**
>
> > 6. Comparison to MAVEN
>
> We add experimental comparison with MAVEN on challenging RWARE and LBF scenarios. We use the mean values over 6 seeds. Our method significantly outperforms this method.
>
> |Method |rware-small-4ag-hard |rware-tiny-4ag-hard|lbf:4s-11x11-3p-2f|
> |-----------------:|---------:|--------:|----------:|
> | SMPE| **6.3**|**20.1**|**98.3**|
> | MAVEN| 1.3|7.8|0.9|
>
> > 7. Real contributions of the paper
>
> Regarding the main contributions, they have been extensively highlighted:
> - in the introduction (Lines 94 left - 69 right)
> - in a series of research questions (Q1-Q8) we address
> - in important results (e.g., see lines 320-326 right: these tasks have been viewed as open challenges by [@])
> - in section 3
>
> Based on the above, the main technical contributions of our paper are the following: We propose the state modelling optimization framework, on top of which we propose the SMPE MARL method. Based on state modelling, each agent of SMPE aims at learning meaningful beliefs about the unobserved global state, from each own perspective, by trying to predict what the other agents observe, and use her belief in order to enhance her own individual policy. To ensure that the beliefs of each agent are informative to her decision making, the framework entails learning of the belief with respect to her policy optimization. Moreover, SMPE introduces the AM filters which aim to filter out redundant global state information which may be detrimental to the inference of the beliefs, and thus of the MARL performance. Our approach introduces two novel loss objectives for learning the beliefs and the AM filters of each agent: (a) a reconstruction loss for learning the AM filters via a self-supervised manner, which ensures that the AM filters identify global state features that can be inferred by the formed agents' belief of the agent, and (b) an RL loss that ensures that the inferred beliefs incorporate information relevant to her policy optimization through backpropagation on the AM filters. Additionally, our method further harnesses the rich state information captured by the agents' beliefs. Each agent uses a count-based intrinsic reward on her own agent belief. This simple exploration schema is proved to be of great importance for joint exploration under partial observability, by encouraging each agent to discover novel observations and also by helping other agents form better explored beliefs about the unobserved global state through the means of an interesting adversarial competition. Remarkably, our experiments validate each of the above conceptual component of our method, and also suggest that SMPE outperforms state-of-the-art methods in challenging, well-established benchmarks.
>
> > 8. Even though the weight converges to including partial non-informative, it can still find a local optimum by policy. More importantly, this suboptimality is difficult to be verified.
>
> Our method controls the contribution of L_rec and L_wcritic to the AM filters through the selection of the hyperparameters: $lr_{wcritic}$ (learning rate in L_wcritic) and $lr_{w}$ (learning rate in L_rec). In practice, we found that a good hyperparameter selection is to set $lr_{wcritic}$ 100 times smaller than $lr_{w}$ (see Table 3, appendix), so that policy optimization does not impede reconstruction. One way to verify if the weight is trained well w.r.t. policy is by looking at the cumulative reward. This is because w affects z and thus both the policy and exploration. In our results (e.g., see Fig. 5) we consider that w is near-optimal w.r.t. policy.

---

### Official Review · Reviewer_W3Cu · 2025-03-23

**Overall Recommendation:** 4

**Summary:**

This paper proposes State Modelling for Policy Enhancement through Exploration, a novel approach to cooperative multi-agent reinforcement learning in partially observable environments without communication. The method enables agents to infer meaningful belief representations about unobservable states through variational inference and self-supervised learning, while filtering out redundant information. The authors claim to enhance agents' policies both explicitly by incorporating these beliefs into policy networks and implicitly through adversarial exploration. Experiments across three benchmarks demonstrate that the proposed consistently outperforms state-of-the-art MARL algorithms, particularly in cooperative tasks that require extensive coordination and exploration.

## update after rebuttal

I have interacted with the authors and my concerns were addressed. I think it is really important to add more details addressing the strength of contribution concerns by the other reviewers for the camera ready version in case of acceptance.

**Claims And Evidence:**

The authors provide theoretical justification (Proposition 2.1) showing that their state modelling objective equals the DecPOMDP objective. They perform extensive experiments on three benchmark environments (MPE, LBF, and RWARE) against multiple baselines with results that agree with their claims.

**Essential References Not Discussed:**

The paper appears to cover the most relevant related works in MARL, agent modelling, and exploration. However, it could benefit from discussing:

1. More recent work on belief representation learning in MARL, such as approaches using transformers or other sequence modeling techniques to handle partial observability.

2. More extensive comparison with world models or model-based MARL approaches

**Experimental Designs Or Analyses:**

The experimental designs and analyses appear sound. The authors use established benchmarks with appropriate configurations and metrics. The ablation studies are well-designed to isolate the contributions of different components.

One concern I have is that some figures show inconclusive results. A more detailed analysis in the style of rliable [1] could be more informative than some subfigures in Fig 3 and 4.
One minor concern is that the evaluation in MPE Spread with increasing number of agents (3, 4, 5, 8) doesn't fully demonstrate the scalability of the approach with even larger numbers of agents.

[1] Agarwal et al, Deep Reinforcement Learning at the Edge of the Statistical Precipice

**Methods And Evaluation Criteria:**

The evaluation metric (unnormalized average episodic reward) is common. The authors report results with confidence intervals averaged over six random seeds, which is sufficient for statistical significance. I appreciate not doing mean and standard error. The authors also properly analyze their method through ablation studies to verify the contribution of each component.

**Other Comments Or Suggestions:**

1. Check Figure 1 and more scalability analysis. Figure 3 and 4 have inconclusive results in some subfigures, it is very clear which ones.

2. Line 262 first column has a typo: “By doing do”

3. The Guan et al. paper is mentioned twice in the references. There is no need or reason.

4. Line 1194 should be *well-entagled* instead

**Other Strengths And Weaknesses:**

The paper is well-written with consistent notations and the clearly stated questions throughout the main body make the paper easy to follow. The formulas are clear and the notation is consistent. Figure 1 is not entirely clear and the arrow convention is not sufficient to distinguish the gradient and the data flow. This work could benefit a lot from more justification on how the baselines are selected, especially the ones that are fairly simpler than the proposed method.

The paper has a clear ablation study but I would like to see more details on how a method with this many components can scale up to more complex settings.

**Questions For Authors:**

1. How does your method scale to more agents?

2. The choice of t_SNE seems somewhat arbitrary? Why choose that visualization over others? What additional observations can you make when using methods like PCA?

3.  Are all the timesteps pooled together for the classification or is the accuracy reported for each timestep separately?

4. Along that line, it would be very interesting if you could show more in-between steps for the visualization. The timestep 45 makes sense but it can also come across as **cherry-picked**

**Relation To Broader Scientific Literature:**

The state modelling framework extends previous agent modelling approaches by learning representations with respect to policy optimization rather than as an auxiliary task. This addresses a limitation in previous works such as "Agent modelling under partial observability for deep reinforcement learning" (Papoudakis et al., 2021) where the aim is to learn a relationship between the trajectory of the controlled agent and the modelled agent. The paper also suggests the use of adversarial methods.

The use of AM filters to handle redundant state information is motivated by findings in "Efficient multi-agent communication via self-supervised information aggregation" (Guan et al., 2022) where the authors aggregate the information through an attention mechanism. Note that the authors mention the same work twice for no reason as two separate entries.

The adversarial exploration schema uses hashing as done in "# exploration: A study of count-based exploration for deep reinforcement learning" (Tang et al., 2017), but applies it in a novel way to the multi-agent setting. I really like the paper’s study of the smoothness of the intrinsic reward smoothness rate.

**Theoretical Claims:**

The authors refer the proof to Proposition 2.1 as “missing proof”. I am not sure what that entails. The proof is sound and follows from the fact that the set of policies encompassed by the state modelling framework includes all policies that could solve the original problem. However, I am not familiar with this methodology.

---

> ### Author Rebuttal · Authors · 2025-03-31
>
> We sincerely appreciate your time and input, along with the positive evaluation. We respond to your comments and questions below.
>
> > 1. A more detailed analysis in the style of rliable [42] could be more informative than some subfigures in Fig 3 and 4.
>
> Thanks for this comment. We will consider using rliable in the camera-ready version.
>
> > 2. Scalability with more agents
>
> Regarding the scalability of our algorithm, Figure 2 shows that it clearly outperforms the other baselines in MPE Spread with 8 agents, albeit by a smaller margin compared to MPE Spread with 3, 4, and 5 agents. Furthermore, additional evaluations in LBF  8s-25x25-8p-5f and 7s-30x30-7p-4f (tasks have been benchmarked by the AAMAS2025 paper [1]) demonstrate that our algorithm achieves superior performance compared to the others.
>
> |Env |SMPE |MAA2C |MAPPO|MASER|EOI|ATM
> |-----------------:|---------:|--------:|----------:|----------:|------:|-----:|
> | 8s-25x25-8p-5f| **0.64 +- 0.12**|0.52 +- 0.24|0.41 +- 0.23|0.01 +- 0| 0.07 +- 0.02 | 0.36 +- 0.28 |
> | 7s-30x30-7p-4f| **0.74 +- 0.02**|0.71 +- 0.02   |0.57 +- 0.03|0.01 +- 0| 0.04 +- 0.01 | 0.59 +- 0.04 |
>
> > 3. Comment on missing related works
>
> Thank you for this comment. Does the reviewer some specific reference in mind? We will be happy to add missing related works.
>
> > 4. Figure 1 is not entirely clear and the arrow convention is not sufficient to distinguish the gradient and the data flow.
>
> Thank you for this comment. Our overview of SMPE^2 and the arrow convention follows a visualization approach similar to that of many well-established MARL papers (see [43], [44], [46]). If the reviewer still has objections about the figure, we are open to discuss possible improvements.
>
> > 5. Justification on how the baselines are selected, especially the ones that are fairly simpler than the proposed method.
>
> Our set of baselines includes some of the most well-known and relevant algorithms to ours. The choice of simpler algorithms was intentional, as MAA2C serves as the backbone of our algorithm, making it a natural baseline. Additionally, while MAPPO is a generalization of the single-agent PPO algorithm, it achieves strong results in our benchmarks, as described in [47].
>
> > 6. The choice of t_SNE seems somewhat arbitrary?
>
> We kindly disagree with this comment. t-SNE is a well established method used widely in the MARL community for embedding visualization (see [43],[44],[45]).
>
> > 7. Are all the timesteps pooled together for the classification or is the accuracy reported for each timestep separately?
>
> Thanks for this comment. Indeed all the timesteps pooled together for the classification.
>
> > 8. The timestep 45 makes sense but it can also come across as cherry-picked
>
> Thanks for this comment. The timestep 45 was not cherry-picked but selected on purpose to be at the end of the trajectory to represent the belief of each agent closer to the solution of the task. Furthermore in the Appendix E.4.8 we provide similar embedding visualizations for the timestep 30.
>
> > 9. Note that the authors mention the same work twice for no reason as two separate entries. Line 262 first column has a typo: “By doing do”. The Guan et al. paper is mentioned twice in the references. Line 1194 should be well-entagled instead
>
> Thanks for your comments. We will fix these minor typos.
>
> **Due to space constraints, below we include responses to some of the questions of the other reviewers.**
>
> > 10. Difference from LIAM, MAVEN and [3]
>
> We respectfully disagree with the reviewer's comment.
>
> - Regarding LIAM, please see lines 669-674. Also, note that LIAM does not use AM for exploration which is a major contribution of our paper.
>
> - Regarding MAVEN, some clear differences are the following:
> --	MAVEN does not account for non-informative state information which can be detrimental to MARL performance.
> --	Our method is not based on committed exploration but on an adversarial competition among the agents, guided by z_i and a count-based method.
> --	Our latent variables are not shared but are unique to each agent.
> --	Our z_i is used as input to each agent's policy, rather than a component of the joint action-value function.
>
> - Regarding [3], first of all, we believe that it is irrelevant as it solely considers single-agent RL, and thus it does not have to deal with MARL challenges, including multi-agent decentralized execution and non-stationarity. Additionally, [3] employs an intrinsic reward based on the prediction error of embeddings of pixel-based states, aiming to identify novel pixels. In stark contrast, we use a count-based hashing method on the belief z_i, which is informative of the unobserved state and policy optimization, ultimately improving joint exploration.
>
> [3] Pathak, Deepak, et al. "Curiosity-driven exploration by self-supervised prediction". 2017.

---

> > ### Comment · Reviewer_W3Cu · 2025-04-08
> >
> > Thank you for your detailed response.
> >
> > > We kindly disagree with this comment. t-SNE is a well established method used widely in the MARL community for embedding visualization (see [43],[44],[45]).
> >
> > This is a minor point but the papers mentioned do not use t-SNE. I am assuming that you are referring to [43][44][45] in your manuscript.
> >
> > > Thanks for this comment. The timestep 45 was not cherry-picked but selected on purpose to be at the end of the trajectory to represent the belief of each agent closer to the solution of the task. Furthermore in the Appendix E.4.8 we provide similar embedding visualizations for the timestep 30.
> >
> > I strongly bleieve the paper would benefit more from a wider sampling of which timesteps to visualize, e.g. including ealier and later steps.

---

> > > ### Author Response · Authors · 2025-04-08
> > >
> > > We would like to thank the reviewer for responding to our rebuttal.
> > >
> > > Regarding the references, the reviewer is right. The references that use t-SNE visualization in MARL are (42; 59; 26) (as we also mention in Line 426 left). We apolozige for this confusion.
> > >
> > > Regarding more timesteps to visualize, we will make sure to include more t-SNE figures in the camera-ready version of our paper.
> > >
> > > Kind regards,
> > >
> > > The authors
> > >
> > > ### References
> > >
> > > [26] Liu, Z., Wan, L., Yang, X., Chen, Z., Chen, X., and Lan, X. Imagine, initialize, and explore: An effective exploration method in multi-agent reinforcement learning. In Proceedings of the AAAI Conference on Artificial Intelligence, volume 38, pp. 17487–17495, 2024
> > >
> > > [42] Papoudakis, G., Christianos, F., and Albrecht, S. Agent modelling under partial observability for deep reinforcement learning. Advances in Neural Information Processing Systems, 34:19210–19222, 2021.
> > >
> > > [59] Xu, P., Zhang, J., Yin, Q., Yu, C., Yang, Y., and Huang, K. Subspace-aware exploration for sparsereward multi-agent tasks. In Proceedings of the AAAI Conference on Artificial Intelligence, volume 37, pp. 11717–11725, 2023.

---

### Decision · Program_Chairs · 2025-05-01

**Decision:**

Accept (poster)

**Comment:**

This paper introduces SMPE^2 a novel framework for cooperative multi-agent reinforcement learning (MARL) in partially observable environments without explicit communication. The core idea involves agents learning meaningful belief representations (z_i) of the unobserved global state from their local observations via variational inference and self-supervised learning. Crucially, the framework incorporates Agent Modellng (AM) filters (w_i) to dynamically weigh information from other agents, aiming to filter out redundant or non-informative features relative to the agent's own policy optimization.

Reviewers generally acknowledged the novelty of the proposed approach, particularly the combination of state modeling with policy-relevance filtering and adversarial exploration mechanism. The paper tackles a significant challenge in MARL w.r.t. improving both individual and joint exploration. The strong empirical results across multiple challenging benchmarks, supported by thorough ablation studies, suggest the method is effective and its components are indeed contributing. The authors actively engaged with reviewer feedback during the rebuttal.